



# Evaluation of 18 satellite- and model-based soil moisture products using *in situ* measurements from 826 sensors

Hylke E. Beck[1,2], Ming Pan[1,2], Diego G. Miralles[3], Rolf H. Reichle[4], Wouter A. Dorigo[5], Sebastian Hahn[5], Justin Sheffield[6,2], Lanka Karthikeyan[7], Gianpaolo Balsamo[8], Robert M. Parinussa[9], Albert I.J.M. van Dijk[10], Jinyang Du[11], John S. Kimball[11], Noemi Vergopolan[1], and Eric F. Wood[1,2]

[1]Department of Civil and Environmental Engineering, Princeton University, Princeton, NJ, USA
[2]Princeton Climate Analytics, Inc., Princeton, NJ, USA
[3]Hydro-Climate Extremes Lab (H-CEL), Ghent University, Ghent, Belgium
[4]Global Modeling and Assimilation Office, NASA Goddard Space Flight Center, Greenbelt, MD, USA
[5]Department of Geodesy and Geoinformation (GEO), Vienna University of Technology, Vienna, Austria
[6]School of Geography and Environmental Science, University of Southampton, Southampton, United Kingdom
[7]Centre of Studies in Resources Engineering, Indian Institute of Technology, Bombay, Powai, Mumbai 400 076, India
[8]European Centre for Medium-Range Weather Forecasts (ECMWF), Reading, UK
[9]School of Geographic Sciences, Nanjing University of Information Science & Technology, Nanjing, Jiangsu, People's Republic of China
[10]Fenner School of Environment and Society, Australian National University, Canberra, Australian Capital Territory, Australia
[11]Numerical Terradynamic Simulation Group, University of Montana, Missoula, MT 59801, USA

**Correspondence:** Hylke E. Beck(hylke.beck@gmail.com)

**Abstract.** Information about the spatiotemporal variability of soil moisture is critical for many purposes, including monitoring of hydrologic extremes, irrigation scheduling, and prediction of agricultural yields. We evaluated the temporal dynamics of 18 state-of-the-art (quasi-)global near-surface soil moisture products, including six based on satellite retrievals, six based on models without satellite data assimilation (referred to hereafter as "open-loop" models), and six based on models that assimilate

5    satellite soil moisture or brightness temperature data. Seven of the products are introduced for the first time in this study: one multi-sensor merged satellite product called MeMo and six estimates from the HBV model with three precipitation inputs (ERA5, IMERG, and MSWEP) and with and without assimilation of SMAPL3E satellite retrievals, respectively. As reference, we used *in situ* soil moisture measurements between 2015 and 2019 at 5-cm depth from 826 sensors, located primarily in the USA and Europe. The 3-hourly Pearson correlation ($R$) was chosen as the primary performance metric. The median $R \pm$ interquartile

10   range across all sites and products in each category was $0.66 \pm 0.30$ for the satellite products, $0.69 \pm 0.25$ for the open-loop models, and $0.72 \pm 0.22$ for the models with satellite data assimilation. The best-to-worst performance ranking of the four single-sensor satellite products was SMAPL3E, SMOS, AMSR2, and ASCAT, with the L-band-based SMAPL3E (median $R$ of 0.72) outperforming the others at 50 % of the sites. Among the two multi-sensor satellite products (MeMo and ESA-CCI), MeMo performed better on average (median $R$ of 0.72 versus 0.67), mainly due to the inclusion of SMAPL3E. The best-to-worst

15   performance ranking of the six open-loop models was HBV-MSWEP, HBV-ERA5, ERA5-Land, HBV-IMERG, VIC-PGF, and GLDAS-Noah. This ranking largely reflects the quality of the precipitation forcing. HBV-MSWEP (median $R$ of 0.78) performed best not just among the open-loop models but among all products. The calibration of HBV improved the median $R$ by +0.12





on average compared to random parameters, highlighting the importance of model calibration. The best-to-worst performance ranking of the six models with satellite data assimilation was HBV-MSWEP+SMAPL3E, HBV-ERA5+SMAPL3E, GLEAM, SMAPL4, HBV-IMERG+SMAPL3E, and ERA5. The assimilation of SMAPL3E retrievals into HBV-IMERG improved the median $R$ by $+0.06$, suggesting that data assimilation yields significant benefits at the global scale.

# 1   Introduction

Accurate and timely information about soil moisture is valuable for many purposes, including drought monitoring, water resources management, irrigation scheduling, prediction of vegetation dynamics and agricultural yields, forecasting floods and heatwaves, and understanding climate change impacts (Wagner et al., 2007; Vereecken et al., 2008; Ochsner et al., 2013; Dorigo and de Jeu, 2016; Brocca et al., 2017; Miralles et al., 2019; Tian et al., 2019; Karthikeyan et al., 2020; Chawla et al., 2020).

Over recent decades, numerous soil moisture products suitable for these purposes have been developed, each with strengths and weaknesses (see Table 1 for a non-exhaustive overview). The products differ in terms of design objective, spatiotemporal resolution and coverage, data sources, algorithm, and latency. They can be broadly classified into three major categories: (i) products directly derived from active- or passive-microwave satellite observations (Zhang and Zhou, 2016; Karthikeyan et al., 2017b), (ii) hydrological or land surface models without satellite data assimilation (referred to hereafter as "open-loop" models;

Cammalleri et al., 2015; Bierkens, 2015; Kauffeldt et al., 2016), and (iii) hydrological or land surface models that assimilate soil moisture retrievals or brightness temperature observations from microwave satellites (Moradkhani, 2008; Liu et al., 2012; Lahoz and De Lannoy, 2014; Reichle et al., 2017).

Numerous studies have evaluated these soil moisture products using *in situ* soil moisture measurements (e.g., Jackson et al., 2010; Bindlish et al., 2018), other independent soil moisture products (e.g., Chen et al., 2018; Dong et al., 2019), remotely-sensed

vegetation greenness data (e.g., Tian et al., 2019), or precipitation data (e.g., Crow et al., 2010; Karthikeyan and Kumar, 2016). Pronounced differences in spatiotemporal dynamics and accuracy were found among the products, even among those derived from the same data source. However, most studies evaluated only one specific product or a small subset ($\leq 3$) of the available products (e.g., Martens et al., 2017; Liu et al., 2019; Zhang et al., 2019b). Additionally, many had a regional (sub-continental) focus (e.g., Albergel et al., 2009; Gruhier et al., 2010; Griesfeller et al., 2016), and thus the extent to which their findings can

be generalized is unclear. Furthermore, several new or recently reprocessed products have not been thoroughly evaluated yet, such as ERA5 (Hersbach et al., 2020), ERA5-Land (C3S, 2019), and ESA-CCI V04.4 (Dorigo et al., 2017). There is also still uncertainty around, for example, the effectiveness of multi-sensor merging techniques (Petropoulos et al., 2015), the impact of model complexity on the accuracy of soil moisture simulations (Fatichi et al., 2016), and the degree to which model deficiencies and precipitation data quality affect the added value of data assimilation (Xia et al., 2019).

Our main objective was to undertake a comprehensive evaluation of 18 state-of-the-art (sub-)daily (quasi-)global near-surface soil moisture products in terms of their temporal dynamics (Section 2.1). Our secondary objective was to introduce seven new soil moisture products (one multi-sensor merged satellite product called MeMo introduced in Section 2.2 and six HBV model-based products introduced in Sections 2.3 and 2.4). As reference for the evaluation, we used *in situ* soil moisture measurements





between 2015 and 2019 from 826 sensors located primarily in the USA and Europe (Section 2.5). We aim to shed light on the advantages and disadvantages of different soil moisture products and on the merit of various technological and methodological innovations by addressing nine key questions:

1. How do the ascending and descending retrievals perform (Section 3.1)?

2. What is the impact of the Soil Wetness Index (SWI) smoothing filter (Section 3.2)?

3. What is the relative performance of the single-sensor satellite products (Section 3.3)?

4. How do the multi-sensor merged satellite products perform (Section 3.4)?

5. What is the relative performance of the open-loop models (Section 3.5)?

6. How do the models with satellite data assimilation perform (Section 3.6)?

7. What is the impact of model calibration (section 3.7)?

8. How do the major product categories compare (Section 3.8)?

9. To what extent are our results generalizable to other regions (Section 3.9)?

## 2   Data and methods

### 2.1   Soil moisture products

We evaluated in total 18 near-surface soil moisture products, including six based on satellite observations, six based on open-loop models, and six based on models that assimilate satellite data (Table 1). The units differed among the products; some are provided in volumetric water content (typically expressed in $m^3$ $m^{-3}$; e.g., ERA5) and others in degree of saturation (typically expressed in %; e.g., ASCAT). We did not harmonize the units among the products, because the Pearson correlation coefficient — the performance metric used in the current study (Section 2.6) — is insensitive to the units. Since the evaluation was performed

at a 3-hourly resolution, we downscaled the two products with a daily temporal resolution (VIC-PGF and GLEAM) to a 3-hourly resolution using nearest neighbor resampling. In contrast to the model products, the satellite products (with the exception of ASCAT) often do not provide retrievals when the soil is frozen or snow-covered (Supplement Fig. S1). To keep the evaluation consistent (Gruber et al., 2020), we discarded the estimates of all 18 products when the near-surface soil temperature was $< 4°C$ and/or the snow depth was $> 1$ mm (both determined using ERA5; Hersbach et al., 2020).

For all satellite products with the exception of MeMo, we also evaluated 3-hourly versions processed using the Soil Wetness Index (SWI) exponential smoothing filter (Wagner et al., 1999; Albergel et al., 2008), which reduces noise and improves the consistency with *in situ* measurements. MeMo was not processed as it was derived from SWI-filtered products. The SWI filter is





defined according to:

$$
\mathrm{SWI}(t) = \frac{\sum\limits_{i} \mathrm{SM}_{\mathrm{sat}}(t_i) e^{-\frac{t-t_i}{T}}}{\sum\limits_{i} e^{-\frac{t-t_i}{T}}}, \tag{1}
$$

where $\mathrm{SM}_{\mathrm{sat}}$ (units depend on the product) is the soil moisture retrieval at time $t_i$, $T$ (days) represents the time lag constant, and $t$ represents the 3-hourly time step. $T$ was set to 5 days for all products, as the performance did not change markedly using different values, as also reported in previous studies (Albergel et al., 2008; Beck et al., 2009; Ford et al., 2014; Pablos et al., 2018). Following Pellarin et al. (2006), the SWI at time $t$ was only calculated if $\geq 1$ retrievals were available in the interval $(t-T,\ t]$ and $\geq 3$ retrievals were available in the interval $[t-3T,\ t-T]$.

## 2.2 Merged soil Moisture (MeMo) product

Merged soil Moisture (MeMo) is a new 3-hourly soil moisture product derived by merging the soil moisture anomalies of three single-sensor passive-microwave satellite products with SWI filter (AMSR2$_{\mathrm{SWI}}$, SMAPL3E$_{\mathrm{SWI}}$, and SMOS$_{\mathrm{SWI}}$; Table 1). MeMo was produced for 2015–2019 (the period with data for all three products) as follows:

1. Three-hourly soil moisture time series of AMSR2$_{\mathrm{SWI}}$, SMAPL3E$_{\mathrm{SWI}}$, SMOS$_{\mathrm{SWI}}$, the active-microwave satellite product ASCAT$_{\mathrm{SWI}}$, and the open-loop model HBV-MSWEP were normalized by subtracting the long-term means and dividing by the long-term standard deviations of the respective products (calculated for the period of overlap).

2. Three-hourly anomalies were calculated for the five products by subtracting their respective seasonal climatologies. The seasonal climatology was calculated by taking the multi-year mean for each day of the year, after which we applied a 30-day central moving mean to eliminate noise. The moving mean was only calculated if $> 21$ days with values were present in the 30-day window. Due to the large number of missing values in winter (Supplement Fig. S1), we were not able to compute the seasonality and, in turn, the anomalies in winter for some satellite products.

3. Time-invariant merging weights for AMSR2$_{\mathrm{SWI}}$, SMAPL3E$_{\mathrm{SWI}}$, and SMOS$_{\mathrm{SWI}}$ were calculated using extended triple collocation (McColl et al., 2014), a technique to estimate Pearson correlation coefficients ($R$) for independent products with respect to an unknown truth. The $R$ values for the respective products were determined using the triplet consisting of the product in question in combination with ASCAT$_{\mathrm{SWI}}$ and HBV-MSWEP, which are independent from each other and from the passive products. The $R$ values were only calculated if $> 200$ coincident anomalies were available. The weights were calculated by squaring the $R$ values.

4. For each 3-hourly time step, we calculated the weighted mean of the available anomalies of AMSR2$_{\mathrm{SWI}}$, SMAPL3E$_{\mathrm{SWI}}$, and SMOS$_{\mathrm{SWI}}$. If only one anomaly was available, this value was used and no averaging was performed. The climatology of SMAPL3E — the best-performing product in our evaluation — was added to the result, to yield the MeMo soil moisture estimates.



## 2.3 HBV hydrological model

Six new 3-hourly soil moisture products were produced using the Hydrologiska Byråns Vattenbalansavdelning (HBV) conceptual hydrological model (Bergström, 1976, 1992) forced with three different precipitation datasets and with and without assimilation of SMAPL3E soil moisture estimates, respectively (Table 1). HBV was selected because of its low complexity, high agility,

computational efficiency, and succesful application used in numerous studies spanning a wide range of climate and physiographic conditions (e.g., Steele-Dunne et al., 2008; Driessen et al., 2010; Beck et al., 2013; Vetter et al., 2015; Jódar et al., 2018). The model has one soil moisture store, two groundwater stores, and 12 free parameters. Among the 12 free parameters, 7 are relevant for simulating soil moisture as they pertain to the snow or soil routines, while 5 are irrelevant for this study as they pertain to runoff generation or deep percolation. The soil moisture store has two inputs (precipitation and snowmelt) and two outputs

(evaporation and recharge). The model was run twice for 2010–2019; the first time to initialize the soil moisture store, and the second time to obtain the final outputs.

HBV requires time series of precipitation, potential evaporation, and air temperature as input. For precipitation, we used three different datasets: (i) the reanalysis ERA5 (hourly $0.28°$ resolution; Hersbach et al., 2020); (ii) the satellite-based IMERG dataset (Late Run V06; 30-minutes $0.1°$ resolution; Huffman et al., 2014, 2018); and (iii) the gauge-, satellite-, and reanalysis-based

MSWEP dataset (V2.4; 3-hourly $0.1°$ resolution; Beck et al., 2017b, 2019b). We calculated 3-hourly accumulations for the ERA5 and IMERG datasets. Daily potential evaporation was estimated using the Hargreaves (1994) equation from daily minimum and maximum air temperature. Temperature estimates were taken from ERA5, downscaled to $0.1°$ and bias-corrected on a monthly basis through an additive approach using the comprehensive station-based WorldClim climatology (V2; 1-km resolution; Fick and Hijmans, 2017). The daily potential evaporation data were downscaled to 3-hourly using nearest neighbour resampling.

We calibrated the 7 relevant parameters of HBV using *in situ* soil moisture measurements between 2010 and 2019 from 177 independent sensors from the International Soil Moisture Network (ISMN) archive that were not used for performance assessment (Section 2.5; Supplement Fig. S2). The parameter space was explored by generating $N = 500$ candidate parameter sets using Latin hypercube sampling (McKay et al., 1979), which splits the parameter space up into $N$ equal intervals and generates parameter sets by sampling each interval once in a random manner. The model was subsequently run for all candidate

parameter sets, after which we selected the parameter set with the best overall performance across the 177 sites (Supplement Table S1). As objective function, we used the median Pearson correlation coefficient ($R$) calculated between 3-hourly *in situ* and simulated soil moisture time series. As forcing, we used the MSWEP precipitation dataset because of its favourable performance in numerous evaluations (e.g., Alijanian et al., 2017; Sahlu et al., 2017; Bai and Liu, 2018; Casson et al., 2018; Beck et al., 2017c, 2019a; Zhang et al., 2019a; Satgé et al., 2019). The calibrated parameter set was used for all HBV runs, including those

forced with ERA5 or IMERG precipitation.

## 2.4 Soil moisture data assimilation

Instantaneous soil moisture retrievals (without SWI filter) from SMAPL3E (Table 1) were assimilated into the HBV model forced with the three above-mentioned precipitation datasets (ERA5, IMERG, and MSWEP). Previous regional studies that





successfully used HBV to assess the value of data assimilation include Parajka et al. (2006), Montero et al. (2016), and Lü et al. (2016). We used the simple Newtonian nudging technique of Houser et al. (1998) that drives the soil moisture state of the model towards the satellite observations. Nudging techniques are computationally efficient and easy to implement, and have therefore been used in several studies (e.g., Brocca et al., 2010b; Dharssi et al., 2011; Capecchi and Brocca, 2014; Laiolo et al., 2016;

Cenci et al., 2016; Martens et al., 2016). For each grid-cell, the soil moisture state of the model was updated when a satellite observation was available according to:

$$\mathrm{SM}_{\mathrm{mod}}^{+}(t) = \mathrm{SM}_{\mathrm{mod}}^{-}(t) + kG\left[\mathrm{SM}_{\mathrm{sat}}^{\mathrm{sc}}(t) - \mathrm{SM}_{\mathrm{mod}}^{-}(t)\right],\qquad(2)$$

where $\mathrm{SM}_{\mathrm{mod}}^{+}$ and $\mathrm{SM}_{\mathrm{mod}}^{-}$ (mm) are the updated and *a priori* soil moisture states of the model, respectively, $\mathrm{SM}_{\mathrm{sat}}^{\mathrm{sc}}$ (mm) are the rescaled satellite observations, and $t$ is the 3-hourly time step. The satellite observations were rescaled to the open-loop model

space using cumulative distribution function (CDF) matching (Reichle and Koster, 2004).

The nudging factor $k$ ($-$) was set to 0.1 as this gave satisfactory results. The gain parameter $G$ ($-$) determines the magnitude of the updates and ranges from 0 to 1. $G$ is generally calculated based on relative quality of the satellite retrievals and the open-loop model. Most previous studies used a spatially and temporally uniform $G$ (e.g., Brocca et al., 2010b; Dharssi et al., 2011; Capecchi and Brocca, 2014; Laiolo et al., 2016; Cenci et al., 2016). Conversely, Martens et al. (2016) used the triple

collocation technique (Scipal et al., 2008) to obtain spatially variable $G$ values. Here we calculated $G$ in a similar fashion according to:

$$G = \frac{R_{\mathrm{sat}}^2}{R_{\mathrm{sat}}^2 + R_{\mathrm{mod}}^2},\qquad(3)$$

where $R_{\mathrm{sat}}$ and $R_{\mathrm{mod}}$ ($-$) are Pearson correlation coefficients with respect to an unknown truth for SMAPL3E and HBV, respectively, calculated using extended triple collocation (Section 2.2). $R_{\mathrm{sat}}$ was determined using 3-hourly anomalies of the

triplet SMAPL3E, ASCAT$_{\mathrm{SWI}}$, and HBV-MSWEP (Table 1) which are based on passive microwaves, active microwaves, and an open-loop model, respectively. $R_{\mathrm{mod}}$ was determined using 3-hourly anomalies of the triplet HBV (forced with either ERA5, IMERG, or MSWEP), ASCAT$_{\mathrm{SWI}}$, and SMAPL3E$_{\mathrm{SWI}}$. The anomalies were calculated by subtracting the seasonal climatologies of the respective products. The seasonal climatologies were determined as described in Section 2.2. The $R_{\mathrm{sat}}$ and $R_{\mathrm{mod}}$ values were only calculated if $> 200$ coincident anomalies were available. The resulting $G$ values vary in space but are constant in time.

**2.5 *In situ* soil moisture measurements**

As reference for the evaluation, we used harmonized and quality-controlled *in situ* volumetric soil moisture measurements ($\mathrm{m^3\,m^{-3}}$) from the ISMN archive (Dorigo et al., 2011, 2013; Appendix Table A1). Similar to numerous previous evaluations (e.g., Albergel et al., 2009; Champagne et al., 2010; Albergel et al., 2012; Wu et al., 2016), we selected measurements from sensors at a depth of 5 cm ($\pm 2$ cm). Since the evaluation was performed at a 3-hourly resolution, the measurements in the ISMN

archive, which have a hourly resolution, were resampled to a 3-hourly resolution. We only used sensors with a 3-hourly record length $> 1$ year (not necessarily consecutive) during the evaluation period from March 31, 2015, to September 16, 2019. We did not average sites with multiple sensors to avoid potentially introducing discontinuities in the time series. In total 826 sensors,

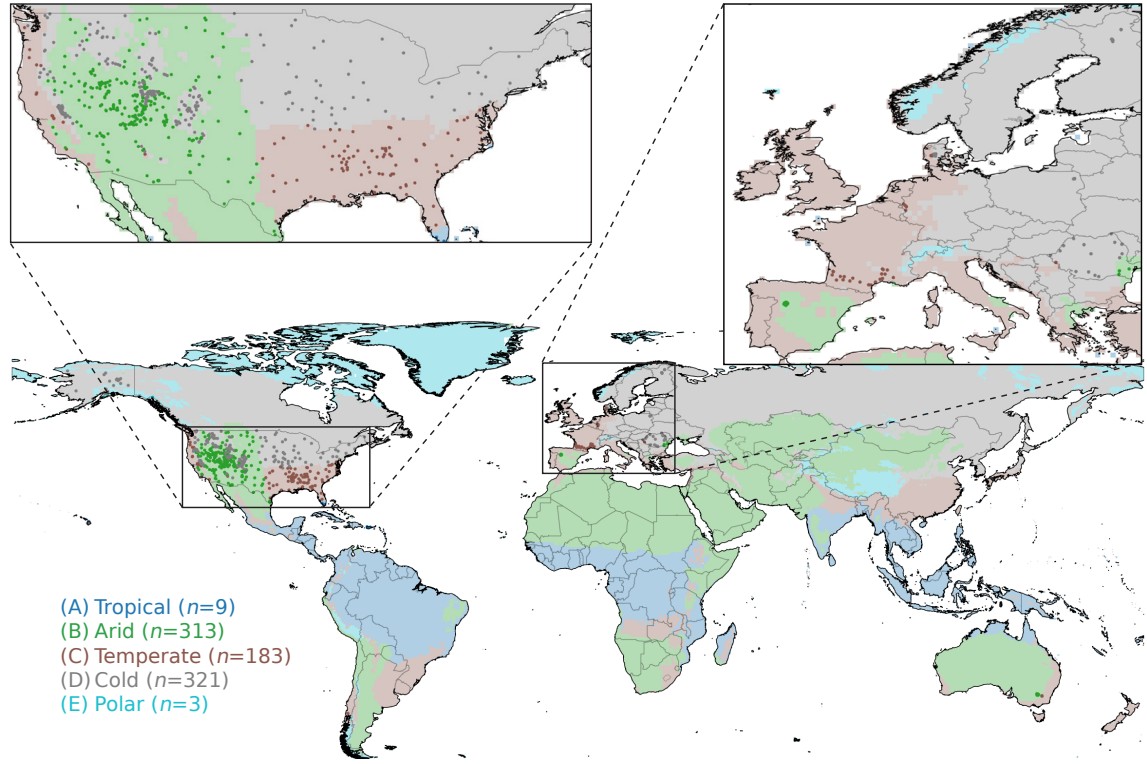

(A) Tropical (*n*=9)
(B) Arid (*n*=313)
(C) Temperate (*n*=183)
(D) Cold (*n*=321)
(E) Polar (*n*=3)

**Figure 1.** Major Köppen-Geiger climate class (Beck et al., 2018) of the 826 sensors used as reference. $n$ denotes the number of sensors in each class.

located in the USA (692), Europe (117), and Australia (17), were available for evaluation (Fig. 1). The median record length was 3.0 years.

## 2.6 Evaluation approach

We evaluated the 18 near-surface soil moisture products (Table 1) for the 4.5-year long period from March 31, 2015 (the date

5 on which SMAP data became available), to September 16, 2019 (the date on which we started processing the products). As performance metric, we used the Pearson correlation coefficient ($R$) calculated between 3-hourly soil moisture time series from the sensor and the product, similar to numerous previous studies (e.g., Karthikeyan et al., 2017a; Al-Yaari et al., 2017; Kim et al., 2018). $R$ measures how well the *in situ* and product time series correspond in terms of temporal variability, and thus evaluates the most important aspect of soil moisture time series for the majority of applications (Entekhabi et al., 2010; Gruber

10 et al., 2020). It is insensitive to systematic differences in mean and variance, which can be substantial due to: (i) the use of different soil property maps as input to the retrieval algorithms and hydrological models (Teuling et al., 2009; Koster et al., 2009); and (ii) the inherent scale discrepancy between *in situ* point measurements and satellite footprints or model grid-cells (Miralles et al., 2010; Crow et al., 2012; Gruber et al., 2020).





Additionally, we calculated Pearson correlation coefficients for the low- and high-frequency fluctuations of the 3-hourly time series ($R_{lo}$ and $R_{hi}$, respectively; Gruber et al., 2020). The low-frequency fluctuations were isolated using a 30-day central moving mean, similar to previous studies (e.g., Albergel et al., 2009; Al-Yaari et al., 2014; Su et al., 2016). The moving mean was calculated only if $> 21$ days with values were present in the 30-day window. The high-frequency fluctuations were isolated

by subtracting the low-frequency fluctuations from the original time series. We discarded the estimates of all products when the near-surface soil temperature was $< 4°C$ and/or the snow depth was $> 1$ mm (both determined using ERA5; Hersbach et al., 2020). For each sensor and product, we only calculated $R$, $R_{hi}$, or $R_{lo}$ values if $> 200$ coincident soil moisture estimates from the sensor and the product were present. Since the spatiotemporal coverage differed among the products (Table 1), the final number of $R$, $R_{hi}$, and $R_{lo}$ values varied depending on the product.

To derive insights into the reasons for the differences in performance, median $R$ values were calculated separately for different Köppen-Geiger climate classes, leaf area index (LAI) values, and topographic slopes. To determine the Köppen-Geiger climate classes, we used the 1-km Köppen-Geiger climate classification map of Beck et al. (2018; Fig. 1), which represents the period 1980–2016. To determine LAI, we used the 1-km Copernicus LAI dataset derived from SPOT-VGT and PROBA-V data (V2; Baret et al., 2016; mean over 1999–2019). To determine the topographic slope, we used the 90-m MERIT DEM (Yamazaki et al.,

2017). To reduce the scale mismatch between point locations and satellite sensor footprints or model grid-cells, we upscaled the Köppen-Geiger, LAI, and topographic slope maps to 0.25° using majority, average, and average resampling, respectively.

## 3   Results and discussion

### 3.1   How do the ascending and descending retrievals perform?

Microwave soil moisture retrievals from ascending and descending overpasses may exhibit performance differences due to

diurnal variations in land surface conditions (Lei et al., 2015) and radio-frequency interference (RFI; Aksoy and Johnson, 2013). Table 2 presents $R$ values for the instantaneous ascending and descending retrievals of the four single-sensor products (AMSR2, ASCAT, SMAPL3E, and SMOS; Table 1). Descending (local night) retrievals were more reliable for the passive microwave-based AMSR2, in agreement with several previous studies (Lei et al., 2015; Griesfeller et al., 2016; Bindlish et al., 2018), and consistent with the notion that soil-vegetation temperature differences during day-time interfere with passive

microwave soil moisture retrieval (Parinussa et al., 2011). Descending (local morning) retrievals were more reliable for the active microwave-based ASCAT (Table 2), in agreement with Lei et al. (2015). The ascending and descending retrievals performed similarly for the passive microwave-based SMAPL3E and SMOS (Table 2). For the remainder of this analysis, we will use only descending retrievals of AMSR2. We did not discard the ascending retrievals of ASCAT as they helped to improve the performance of ASCAT$_{SWI}$.



**Figure 2.** (a) Performance of the soil moisture products in terms of 3-hourly Pearson correlation ($R$). The products were sorted in ascending order of median $R$. Outliers are not shown. The number above the median line in each box represents the number of sites with $R$ values and the number below the median line represents the median $R$ value. Also shown are median $R$ values for different (b) Köppen-Geiger climate classes, (c) LAI values, and (d) topographic slopes.



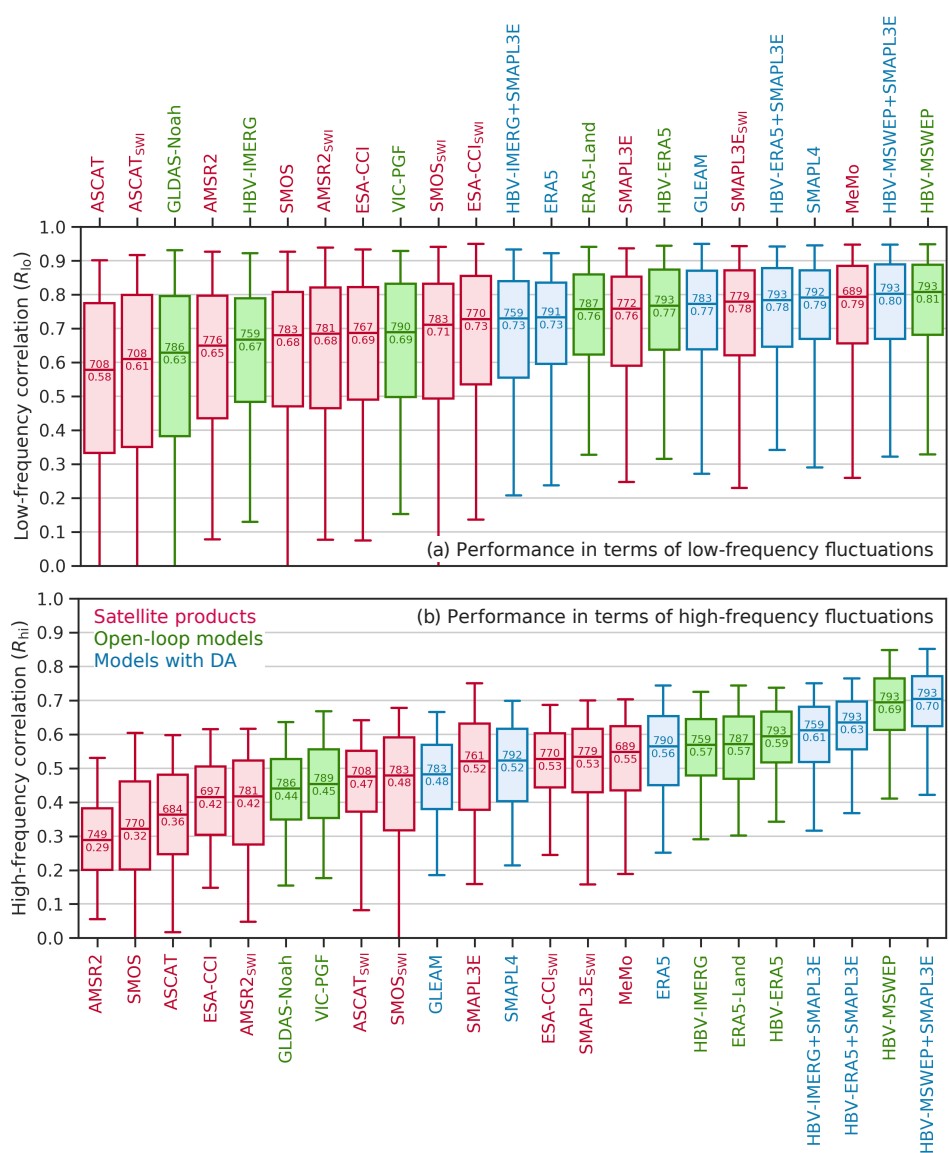

**Figure 3.** Performance of the soil moisture products in terms of 3-hourly Pearson correlation for (a) low-frequency fluctuations ($R_{\text{lo}}$) and (b) high-frequency fluctuations ($R_{\text{hi}}$). The products were sorted in ascending order of the median. The number above the median line in each box represents the number of sites with $R_{\text{lo}}$ or $R_{\text{hi}}$ values and the number below the median line represents the median $R_{\text{lo}}$ or $R_{\text{hi}}$ value. Outliers are not shown.





## 3.2 What is the impact of the Soil Wetness Index (SWI) smoothing filter?

The application of the SWI filter resulted in higher median $R$, $R_{hi}$, and $R_{lo}$ values for all satellite products (Figs. 2a and 3; Table 1). The median $R$ improvement was $+0.12$ for AMSR2, $+0.10$ for ASCAT, $+0.07$ for SMAPL3E, $+0.17$ for SMOS, and $+0.11$ for ESA-CCI (Fig. 2a). The improvements are probably mainly because the SWI filter reduces the impact of random

errors and potential differences between ascending and descending overpasses (Su et al., 2015; Bogoslovskiy et al., 2015). Additionally, since the SWI filter simulates the slower variability of soil moisture at deeper layers (Wagner et al., 1999; Albergel et al., 2008; Brocca et al., 2010a), it improves the consistency between the *in situ* measurements at 5-cm depth and the microwave signals, which often have a penetration depth of just 1–2 cm depending on the observation frequency and the land surface conditions (Long and Ulaby, 2015; Shellito et al., 2016a; Rondinelli et al., 2015; Lv et al., 2018). Our results suggests that

previous near-surface soil moisture product assessments (e.g., Zhang et al., 2017; Karthikeyan et al., 2017a; Cui et al., 2018; Al-Yaari et al., 2019; Ma et al., 2019), which generally did not use smoothing filters, may have underestimated the true skill of the products.

## 3.3 What is the relative performance of the single-sensor satellite products?

Among the four single-sensor products with SWI filter (AMSR2$_{SWI}$, ASCAT$_{SWI}$, SMAPL3E$_{SWI}$, and SMOS$_{SWI}$; Table 1),

SMAPL3E$_{SWI}$ performed best in terms of median $R$, $R_{lo}$, and $R_{hi}$ by a wide margin (Figs. 2a and 3), in agreement with previous studies using triple collocation (Chen et al., 2018) and *in situ* measurements from the USA (Karthikeyan et al., 2017a; Zhang et al., 2017; Cui et al., 2018; Al-Yaari et al., 2019), the Tibetan Plateau (Chen et al., 2017), the Iberian Peninsula (Cui et al., 2018), and across the globe (Al-Yaari et al., 2017; Kim et al., 2018; Ma et al., 2019). The good performance of SMAPL3E$_{SWI}$ is likely attributable to the deeper ground penetration of L-band signals (Lv et al., 2018), the sensor's higher radiometric

accuracy (Entekhabi et al., 2010), and the application of an RFI mitigation algorithm (Piepmeier et al., 2014). SMOS$_{SWI}$ is also an L-band product, while the AMSR2$_{SWI}$ product used here was derived from X-band observations, which have a shallower penetration depth (Long and Ulaby, 2015). Both AMSR2$_{SWI}$ and SMOS$_{SWI}$ are more vulnerable to RFI, which may have reduced their overall performance (Njoku et al., 2005; Oliva et al., 2012). The active microwave-based ASCAT$_{SWI}$ performed significantly better in terms of high-frequency than low-frequency fluctuations (Fig. 3), likely due to the presence of seasonal

vegetation-related biases (Wagner et al., 2013). ASCAT$_{SWI}$ showed a relatively small spread in $R_{hi}$ values (Fig. 3b), although it showed the largest spread in $R$ and $R_{lo}$ values not just among the single-sensor products but among all products (Figs. 2a and 3a).

All single-sensor satellite products achieved lower $R$ values in cold climates (Figs. 1 and 2b), in agreement with other global evaluations using ISMN data (Kim et al., 2018; Al-Yaari et al., 2019; Zhang et al., 2019b; Ma et al., 2019), and previously

attributed to the confounding influence of dense vegetation cover (de Rosnay et al., 2006; Gruhier et al., 2008; Dorigo et al., 2010), highly organic soils (Zhang et al., 2019b), and standing water (Ye et al., 2015; Du et al., 2018) on soil moisture retrievals. However, since the models also tend to exhibit lower $R$ values in cold regions (Fig. 2b), it could also be that the *in situ* measurements are of lower quality and/or that our procedure to screen for frozen or snow-covered soils is imperfect. AMSR2





and particularly AMSR2$_{\text{SWI}}$ performed noticeably better in terms of $R$ in arid climates (Figs. 1 and 2b), as reported in previous studies (Wu et al., 2016; Cho et al., 2017), and likely due to the availability of coincident Ka-band brightness temperature observations which are used as input to the LPRM retrieval algorithm (Parinussa et al., 2011). AMSR2 and SMOS (with and without SWI filter) showed markedly lower $R$ values for sites with mean leaf area index $> 2 \text{ m}^2 \text{ m}^{-2}$ (Fig. 2c), confirming that

their retrievals are affected by dense vegetation cover (Al-Yaari et al., 2014; Wu et al., 2016; Cui et al., 2018). Most satellite products performed worse in terms of $R$ in areas of steep terrain (Fig. 2d), consistent with previous evaluations (Paulik et al., 2014; Karthikeyan et al., 2017a; Ma et al., 2019), and attributed to the confounding effects of relief on the upwelling microwave brightness temperature observed by the radiometer (Mialon et al., 2008; Pulvirenti et al., 2011; Guo et al., 2011).

### 3.4    How do the multi-sensor merged satellite products perform?

The multi-sensor merged product MeMo (based on AMSR2$_{\text{SWI}}$, SMAPL3E$_{\text{SWI}}$, and SMOS$_{\text{SWI}}$) performed better than the four single-sensor products for all three metrics ($R$, $R_{\text{lo}}$, and $R_{\text{hi}}$; Figs. 2a and 3; Table 1). These results highlight the value of multi-sensor merging techniques, in line with prior studies that merged satellite retrievals (Gruber et al., 2017; Kim et al., 2018), model outputs (Guo et al., 2007; Liu and Xie, 2013; Cammalleri et al., 2015), and satellite retrievals with model outputs (Yilmaz et al., 2012; Anderson et al., 2012; Tobin et al., 2019; Vergopolan et al., 2020). However, MeMo performed only marginally

better in terms of $R$ than the best-performing single-sensor product SMAPL3E$_{\text{SWI}}$ (which was incorporated in MeMo; Fig. 2a). The most likely reason for this is probably that since all products incorporated in MeMo are based on passive-microwave remote sensing, their errors may to a certain degree be cross-correlated and hence may not fully cancel each other out (Yilmaz and Crow, 2014).

Additionally, MeMo performed better than the multi-sensor merged product ESA-CCI$_{\text{SWI}}$ (based on AMSR2, ASCAT, and

SMOS) for all three metrics (Figs. 2a and 3). MeMo performed better in terms of $R$ at 68 % of the sites, and performed particularly well across the central Rocky Mountains, although ESA-CCI$_{\text{SWI}}$ performed better in eastern Europe (Fig. 4). The two products performed similarly in terms of high-frequency fluctuations (median $R_{\text{hi}}$ of 0.55 for MeMo versus 0.53 for ESA-CCI$_{\text{SWI}}$; Fig. 3b). The better overall performance of MeMo compared to ESA-CCI$_{\text{SWI}}$ (Figs. 2a, 3, and 4) is probably due to two factors. First, ESA-CCI$_{\text{SWI}}$ incorporates ASCAT, which performed less well in the present evaluation, whereas

MeMo incorporates SMAPL3E$_{\text{SWI}}$, which performed best among the single-sensor products (Figs. 2a and 3). The median $R$ of MeMo dropped by 0.04 after we excluded SMAPL3E$_{\text{SWI}}$ (data not shown), which supports this explanation. The next version of ESA-CCI (V5) is anticipated to incorporate SMAP soil moisture estimates, and is therefore expected to perform better (Gruber et al., 2019). Secondly, MeMo merges soil moisture estimates from multiple sensors each day, whereas ESA-CCI$_{\text{SWI}}$ uses only the soil moisture estimate from the 'best' sensor each day, resulting in a loss of information.

### 3.5    What is the relative performance of the open-loop models?

The ranking of the six open-loop models in terms of median $R$ (from best to worst) was (i) HBV-MSWEP, (ii) HBV-ERA5, (iii) ERA5-Land, (iv) HBV-IMERG, (v) VIC-PGF, and (vi) GLDAS-Noah (Fig. 2a; Table 1). The models were forced with precipitation from, respectively: (i) the gauge-, satellite-, and reanalysis-based MSWEP V2.4 (Beck et al., 2017b, 2019b),



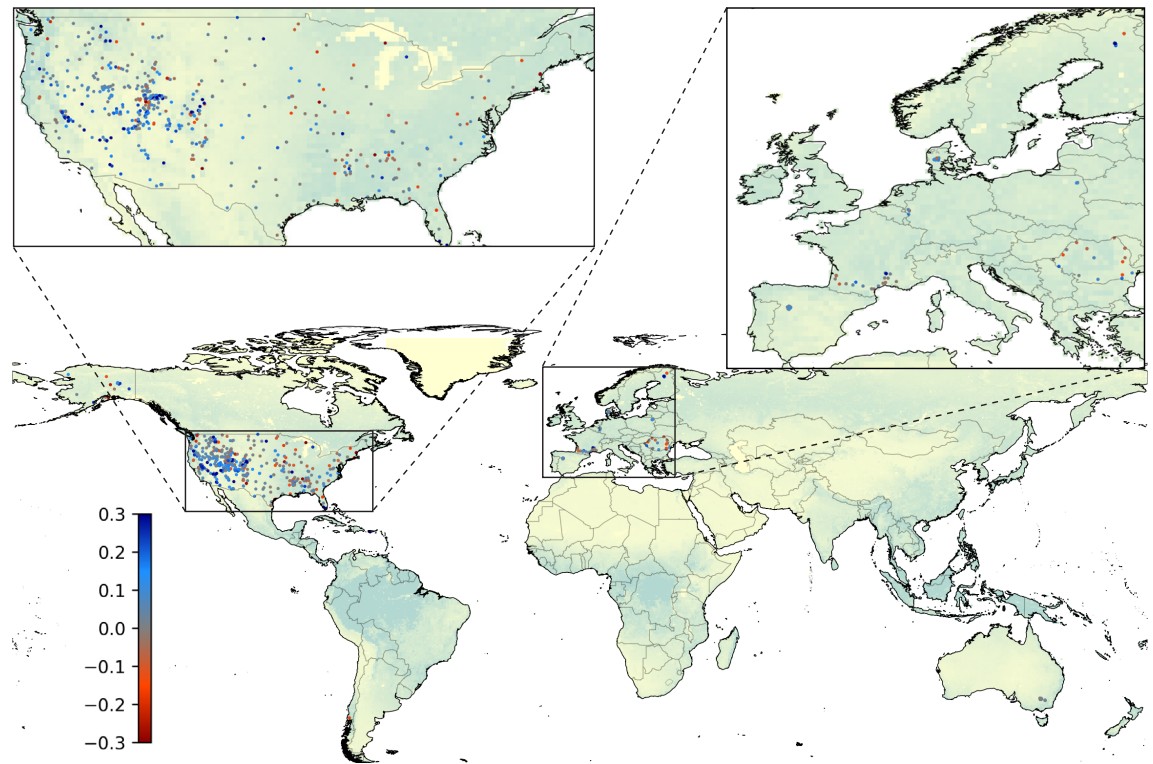

**Figure 4.** Three-hourly Pearson correlations ($R$) obtained by MeMo minus those obtained by ESA-CCI. Blue indicates that MeMo performs better, whereas red indicates that ESA-CCI performs better. A map of long-term mean LAI (Baret et al., 2016) is plotted in the background.

(ii) and (iii) the ERA5 reanalysis (Hersbach et al., 2020), (iv) the satellite-based IMERGHHE V06 (Huffman et al., 2014, 2018), (v) the gauge- and reanalysis-based PGF (Sheffield et al., 2006), and (vi) the gauge- and satellite-based GPCP V1.3 Daily Analysis (Huffman et al., 2001). This order matches the overall performance ranking of precipitation datasets in a comprehensive evaluation over the conterminous USA carried out by Beck et al. (2019a). Furthermore, the performance of HBV-ERA5 did not

5 depend on the terrain slope, while HBV-IMERG performed worse in steep terrain (Fig. 2d), which is also consistent with the evaluation of Beck et al. (2019a). HBV-IMERG performed worse for low-frequency than for high-frequency fluctuations (Fig. 3), which likely reflects the presence of seasonal biases in IMERG (Beck et al., 2017c; Wang and Yong, 2020). Overall, these results confirm that precipitation is by far the most important determinant of soil moisture simulation performance (Gottschalck et al., 2005; Liu et al., 2011; Beck et al., 2017c; Dong et al., 2019). The superior performance of MSWEP is primarily attributable to

10 the daily gauge corrections (Beck et al., 2019b).

Among the three soil moisture products derived from ERA5 precipitation (ERA5, ERA5-Land, and HBV-ERA5), and among the three products forced with daily gauge-corrected precipitation (GLEAM, HBV-MSWEP+SMAPL3E, and SMAPL4; Table 1), the ones based on HBV performed better overall in terms of all three metrics ($R$, $R_{lo}$, and $R_{hi}$; Figs. 2a and 3). This demonstrates that soil moisture estimates from complex, data-intensive models (H-TESSEL underlying ERA5 and ERA5-Land, GLEAM,





and the Catchment model underlying SMAPL4) are not necessarily more accurate than those from relatively simple, calibrated models (HBV). This is in line with several previous multi-model evaluations focusing on soil moisture (e.g., Guswa et al., 2002; Cammalleri et al., 2015; Orth et al., 2015), the surface energy balance (e.g., Best et al., 2015), evaporation (e.g., McCabe et al., 2016), runoff (e.g., Beck et al., 2017a), and river discharge (e.g., Gharari et al., 2020).

### 3.6   How do the models with satellite data assimilation perform?

The performance ranking of the models with satellite data assimilation in terms of median $R$ (from best to worst) was HBV-MSWEP+SMAPL3E, HBV-ERA5+SMAPL3E, GLEAM, SMAPL4, HBV-IMERG+SMAPL3E, and ERA5 (Fig. 2a; Table 1). The assimilation of SMAPL3E retrievals resulted in a substantial improvement in median $R$ of $+0.06$ for HBV-IMERG, a minor improvement of $+0.01$ for HBV-ERA5, and no change for HBV-MSWEP (Fig. 2a). Improvements in $R$ were obtained for 90 %,
65 %, and 56 % of the sites for HBV-IMERG, HBV-ERA5, and HBV-MSWEP, respectively (Fig. 5). These results suggest that data assimilation provides greater benefits when the precipitation forcing is more uncertain (Beck et al., 2019a). Since rain gauge observations are not available over the large majority of the globe (Kidd et al., 2017), we expect data assimilation to provide significant added value at the global scale, as also concluded by Bolten et al. (2010), Dong et al. (2019), and Tian et al. (2019). The lack of improvement for HBV-ERA5+SMAPL3E and HBV-MSWEP+SMAPL3E suggests that the gain
parameter $G$ (Eq. 3), which quantifies the relative quality of the satellite and model soil moisture estimates, can be refined further.

The ERA5 reanalysis, which assimilates ASCAT soil moisture (Hersbach et al., 2020), obtained a lower overall performance (median $R = 0.68$) than the open-loop models ERA5-Land (median $R = 0.72$) and HBV-ERA5 (median $R = 0.74$), which were both forced with ERA5 precipitation (Fig. 2a). This suggests that assimilating satellite soil moisture estimates (ERA5) was less
beneficial than either increasing the model resolution (ERA5-Land) or improving the model efficiency (HBV). In line with these results, Muñoz Sabater et al. (2019) found that the joint assimilation of ASCAT soil moisture retrievals and SMOS brightness temperatures into an experimental version of the Integrated Forecast System (IFS) model underlying ERA5 did not improve the soil moisture simulations. They attributed this to the adverse impact of simultaneously assimilated screen-level temperature and relative humidity observations on the soil moisture estimates.

In line with our results for HBV-MSWEP+SMAPL3E, Kumar et al. (2014) did not obtain improved soil moisture estimates after the assimilation of ESA-CCI and AMSR-E retrievals into Noah forced with highly accurate NLDAS2 meteorological data for the conterminous USA. Conversely, several other studies obtained substantial performance improvements after data assimilation despite the use of high-quality precipitation forcings (Liu et al., 2011; Koster et al., 2018; Tian et al., 2019). We suspect that this discrepancy might reflect the lower performance of their open-loop models compared to ours. Using different
(but overlapping) *in situ* datasets, Koster et al. (2018) and Tian et al. (2019) obtained mean daily open-loop $R$ values of 0.64 and 0.59, respectively, while we obtained a mean daily open-loop $R$ of 0.75 (slightly lower than the 3-hourly median value shown in Fig. 2a). Overall, it appears that the benefits of data assimilation are greater for models that exhibit structural or parameterization deficiencies.

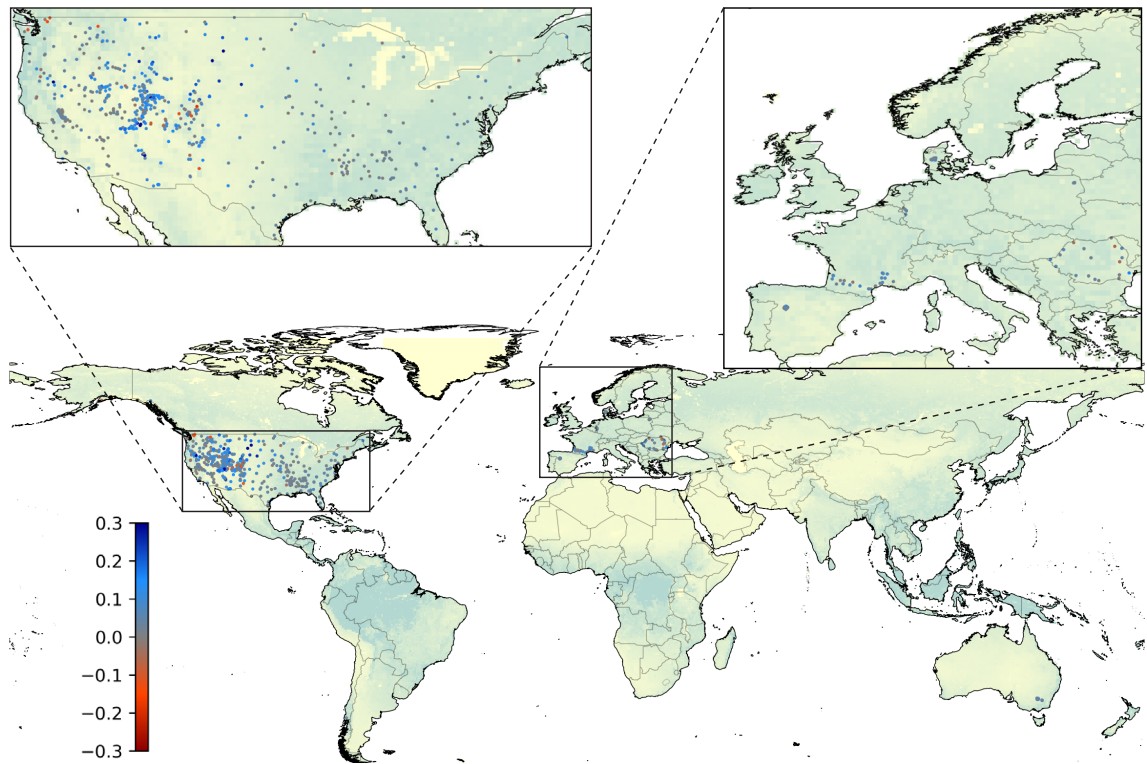

**Figure 5.** Three-hourly Pearson correlations ($R$) obtained by HBV-IMERG+SMAPL3E minus those obtained by HBV-IMERG. Blue indicates improved performance after data assimilation, whereas red indicates degraded performance after data assimilation. The sites in Finland are not shown because IMERG does not cover high latitudes. A map of long-term mean LAI (Baret et al., 2016) is plotted in the background.

### 3.7 What is the impact of model calibration?

Among the models evaluated in this study, only HBV and the Catchment model underlying SMAPL4 have been calibrated, although only a single parameter out of more than 100 was calibrated for the Catchment model (Reichle et al., 2019b). HBV-MSWEP with calibrated parameters obtained a median $R$ of 0.78 (Fig. 2a), whereas HBV-MSWEP with randomly generated

5  parameters obtained a mean median $R$ of 0.66 (standard deviation 0.07; data not shown). The calibration thus resulted in a mean increase in median $R$ of +0.12, which represents a substantial improvement in performance. These results are in line with previous studies calibrating different models using soil moisture from *in situ* sensors (e.g., Koren et al., 2008; Shellito et al., 2016b; Thorstensen et al., 2016; Reichle et al., 2019b) or remote sensing (e.g., Zhang et al., 2011; Wanders et al., 2014; López López et al., 2016; Koster et al., 2018).

10  The mean improvement in median $R$ obtained for HBV-MSWEP after calibration (+0.12) was double the improvement obtained for HBV-IMERG after satellite data assimilation (+0.06; Fig. 2a; Section 3.6), which suggests that model calibration is more beneficial overall than data assimilation. Additionally, model calibration is likely to benefit regions with both sparse and





dense rain gauge networks, whereas data assimilation mainly benefits regions with sparse rain gauge networks (Section 3.6). Conversely, only data assimilation is capable of ameliorating potential deficiencies in the meteorological forcing data (e.g., undetected precipitation).

Our calibration approach was relatively simple and yielded only a single spatially uniform parameter set (Section 2.3). Previous
studies focusing on runoff have demonstrated the value of more sophisticated calibration approaches yielding ensembles of parameters that vary according to climate and landscape characteristics (Beck et al., 2016, in review). Whether these approaches have value for soil moisture estimation as well warrants further investigation. It should be noted, however, that many current models have rigid structures, insufficient free parameters, and/or a high computational cost which makes them less amenable to calibration (Mendoza et al., 2015). Moreover, the validity of calibrated parameters may be compromised when the model is
subjected to climate conditions it has never experienced before (Knutti, 2008). Care should also be taken that calibration of one aspect of the model does not degrade another aspect and that we get "the right answers for the right reasons" (Kirchner, 2006). the model can be extrapolated beyond the range of where they are evaluated.

## 3.8   How do the major product categories compare?

The median $R \pm$ interquartile range across all sites and products in each category was $0.53 \pm 0.32$ for the satellite soil moisture
products without SWI filter, $0.66 \pm 0.30$ for the satellite soil moisture products with SWI filter including MeMo, $0.69 \pm 0.25$ for the open-loop models, and $0.72 \pm 0.22$ for the models with satellite data assimilation (Fig. 2a; Table 1). The satellite products thus provided the least reliable soil moisture estimates and exhibited the largest regional performance differences on average, whereas the models with satellite data assimilation provided the most reliable soil moisture estimates and exhibited the smallest regional performance differences on average. Our performance ranking of the major product categories is consistent with
previous studies for the conterminous USA (Liu et al., 2011; Kumar et al., 2014; Fang et al., 2016; Dong et al., 2020), Europe (Naz et al., 2019), and the globe (Albergel et al., 2012; Tian et al., 2019; Dong et al., 2019). It should be kept in mind, however, that these studies, including the present one, used *in situ* soil moisture measurements from regions with dense rain gauge networks, and hence likely overestimate model performance (Dong et al., 2019).

The large spread in performance across the satellite products reflects the large number of factors that affect soil moisture
retrieval, including vegetation cover, surface roughness, soil texture, diurnal variations in land surface conditions, and RFI, among others (Zhang and Zhou, 2016; Karthikeyan et al., 2017b). The spread in performance across the open-loop models is lower as it depends primarily on the precipitation data quality, which, in turn, depends mostly on a combination of gauge network density and prevailing precipitation type (convective versus stratiform; Gottschalck et al., 2005; Liu et al., 2011; Beck et al., 2017c; Dong et al., 2019). The smaller spread in performance across the models with satellite data assimilation is due to
the fact that individual errors in satellite retrievals and model estimates are cancelled out, to a certain degree, when they are combined, confirming the effectiveness of the data assimilation procedures (Moradkhani, 2008; Liu et al., 2012; Reichle et al., 2017).





### 3.9 To what extent are our results generalizable to other regions?

The large majority (98 %) of the *in situ* soil moisture measurements used as reference in the current study were from the USA and Europe (Fig. 1). We speculate that our results for the models (with and without data assimilation; Figs. 2, 3, and 5) apply to other regions with dense rain gauge networks and broadly similar climates (e.g., parts of China and Australia, and other parts
of Europe; Kidd et al., 2017). In sparsely gauged areas the model products based on precipitation forcings that incorporate daily gauge observations (GLEAM, HBV-MSWEP, HBV-MSWEP+SMAPL3E, and SMAPL4; Table 1) will inevitably exhibit reduced performance. In convection-dominated regions models driven by precipitation from satellite datasets such as IMERG may well outperform those driven by precipitation from reanalyses such as ERA5 (Massari et al., 2017; Beck et al., 2017c, 2019b). Conversely, in mountainous and snow-dominated regions models driven by precipitation from reanalyses are likely to
outperform those driven by precipitation from satellites (Ebert et al., 2007; Beck et al., 2019b, a).

Our results for the satellite soil moisture products may be less generalizable, given the large spread in performance across different regions and products revealed in the current study (Figs. 2 and 3) and in previous quasi-global studies using triple collocation (Al-Yaari et al., 2014; Chen et al., 2018; Miyaoka et al., 2017). Outside developed regions we expect the lower prevalence of RFI to lead to more reliable retrievals for those satellite products susceptible to it (Njoku et al., 2005; Oliva et al.,
2012; Aksoy and Johnson, 2013; Ticconi et al., 2017). At low latitudes the lower satellite revisit frequency will inevitably increase the sampling uncertainty and reduce the overall value of satellite products relative to models. In tropical forest regions passive products often do not provide soil moisture retrievals, and when they do, the retrievals are typically less reliable than those from active products due to the dense vegetation cover (Al-Yaari et al., 2014; Chen et al., 2018; Miyaoka et al., 2017; Kim et al., 2018). Shedding more light on the strengths and weaknesses of soil moisture products in regions without dense
measurement networks — for example using independent soil moisture products (Chen et al., 2018; Dong et al., 2019) or by expanding measurement networks (Kang et al., 2016; Singh et al., 2019) — should be a key priority for future research (Ochsner et al., 2013; Myeni et al., 2019).

## 4   Conclusions

To shed light on the advantages and disadvantages of different soil moisture products and on the merit of various technological
and methodological innovations, we evaluated 18 state-of-the-art (sub-)daily (quasi-)global near-surface soil moisture products using *in situ* measurements from 826 sensors located primarily in the USA and Europe. Our main findings related to the nine questions posed in the introduction can be summarized as follows:

1. Local night retrievals from descending overpasses were more reliable overall for AMSR2, whereas local morning retrievals from descending overpasses were more reliable overall for ASCAT. The ascending and descending retrievals of SMAPL3E
and SMOS performed similarly.





2. Application of the SWI smoothing filter resulted in improved performance for all satellite products. Previous near-surface soil moisture product assessments generally did not apply smoothing filters and therefore may have underestimated the true skill of the products.

3. SMAPL3E$_{SWI}$ performed best overall among the four single-sensor satellite products with SWI filter. ASCAT$_{SWI}$ performed markedly better in terms of high-frequency than low-frequency fluctuations. All satellite products tended to perform worse in cold climates.

4. The multi-sensor merged satellite product MeMo performed best among the satellite products, highlighting the value of multi-sensor merging techniques. MeMo also outperformed the multi-sensor merged satellite product ESA-CCI$_{SWI}$, likely due to the inclusion of SMAPL3E$_{SWI}$.

5. The performance of the open-loop models depended primarily on the precipitation data quality. The superior performance of HBV-MSWEP is due to the calibration of HBV and the daily gauge corrections of MSWEP. Soil moisture simulation performance did not improve with model complexity.

6. In the absence of model structural or parameterization deficiencies, satellite data assimilation yields substantial performance improvements mainly when the precipitation forcing is of relatively low quality. This suggests that data assimilation provides significant benefits at the global scale.

7. The calibration of HBV against *in situ* soil moisture measurements resulted in substantial performance improvements. The improvement due to model calibration tends to exceed the improvement due to satellite data assimilation and is not limited to regions of low quality precipitation.

8. The satellite products provided the least reliable soil moisture estimates and exhibited the largest regional performance differences on average, whereas the models with satellite data assimilation provided the most reliable soil moisture estimates and exhibited the smallest regional performance differences on average.

9. We speculate that our results for the models (with and without data assimilation) apply to other regions with dense rain gauge networks and broadly similar climates. Our results for the satellite products may be less generalizable due to the large number of factors that affect retrievals.

**Appendix: *In situ* soil moisture measurement networks**

Table A1 lists the measurement networks part of the ISMN archive from which we have used *in situ* soil moisture data.

*Author contributions.* H.E.B. conceived, designed, and performed the analysis and took the lead in writing the paper. E.F.W. was responsible for funding acquisition. All co-authors provided critical feedback and contributed to the writing.



*Competing interests.* The authors declare no competing interests.

*Acknowledgements.* We are grateful to the numerous contributors to the ISMN archive. The soil moisture product developers are thanked for producing and making available their products. Hylke E. Beck and Ming Pan were supported through IPA support from the U.S. Army Corps of Engineers' International Center for Integrated Water Resources Management (ICIWaRM), under the auspices of UNESCO. Robert M. Parinussa was funded by National Natural Science Foundation of China grant 41850410492. Noemi Vergopolan was in part supported by NASA Soil Moisture Cal/Val Activities (NNX14AH92G). Noemi Vergopolan and Ming Pan were in part supported by NOAA's "Modernizing Observation Operator and Error Assessment for Assimilating In-situ and Remotely Sensed Snow/Soil Moisture Measurements into NWM" project (NA19OAR4590199). Rolf H. Reichle and John S. Kimball were supported by the NASA SMAP Science Team. Diego G. Miralles acknowledges support from the European Research Council (ERC) DRY-2-DRY project (715254) and the Belgian Science Policy Office (BELSPO) STEREO III ALBERI project (SR/00/373). Wouter A. Dorigo acknowledges support from the ESA CCI Programme and ESA's IDEAS+ and QA4EO projects.





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

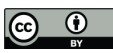



**Table 1.** The 18 soil moisture products evaluated in this study. For the single-sensor satellite products, the spatial resolution represents the footprint size and the temporal resolution the average revisit time. Acronyms: A = ascending; D = descending; PMW = passive microwave; AMW = active microwave; $P$ = precipitation; DA = data assimilation.

| Acronym | Details | Spatial resolution | Temporal resolution | Temporal coverage | Latency | Reference(s) |
|---|---|---|---|---|---|---|
| *Satellite products* | | | | | | |
| AMSR2[a] | AMSR2/GCOM-W1 LPRM L3 V001 (soil_moisture_x); single-sensor PMW product; only D passes | ~47 km | 1–3 days | 2012–present | Several hours | Parinussa et al. (2015) |
| ASCAT[a] | Combination of H115 and H116; single-sensor AMW product; A and D passes | ~30 km | 1–2 days | 2007–present | Several hours | Wagner et al. (2013); H SAF (2019a, b) |
| SMAPL3E[a] | SPL3SMP_E.003 L3 Enhanced Radiometer EASE-Grid V3; single-sensor PMW product; A and D passes | ~30 km | 1–3 days | 2015–present | Several hours | Entekhabi et al. (2010); Chan et al. (2018); O'Neill et al. (2019) |
| SMOS[a] | L2 User Data Product (MIR_SMUDP2) V650; single-sensor PMW product; A and D passes | ~40 km | 1–3 days | 2010–present | Several hours | Kerr et al. (2012) |
| ESA-CCI[a] | ESA-CCI SM V04.4 COMBINED; multi-sensor merged AMW- and PMW-based product derived from AMSR2, ASCAT, and SMOS | 0.25° | Daily | 1978–2018 | About a year | Dorigo et al. (2017); Gruber et al. (2019) |
| MeMo | Multi-sensor merged PMW product derived from AMSR2, SMAPL3E, and SMOS with SWI filter | 0.1° | 3-hourly | 2015–present | Several hours | This study (Section 2.2) |
| *Open-loop models (i.e., without data assimilation)* | | | | | | |
| ERA5-Land | Volumetric soil water layer 1 (0–7 cm); H-TESSEL model; forced with ERA5 $P$ (Hersbach et al., 2020) | 0.1° | Hourly | 1979–2020 | Several months | C3S (2019) |
| GLDAS-Noah | GLDAS_NOAH025_3H.2.1 (SoilMoi0_10cm_inst) forced with GPCP V1.3 Daily Analysis $P$ (Huffman et al., 2001) | 0.25° | 3-hourly | 1948–2020 | 2–3 months | Rodell et al. (2004); Rui et al. (2020) |
| HBV-ERA5 | HBV forced with ERA5 $P$ (Hersbach et al., 2020) | 0.28° | 3-hourly | 1979–2020 | Several months | This study (Section 2.3) |
| HBV-IMERG | HBV forced with IMERGHHE V06 $P$ (Huffman et al., 2014, 2018) | 0.1° | 3-hourly | 2000–present | Several hours | This study (Section 2.3) |
| HBV-MSWEP | HBV forced with MSWEP V2.4 $P$ (Beck et al., 2019b) | 0.1° | 3-hourly | 2000–present | Several hours[b] | This study (Section 2.3) |
| VIC-PGF | Layer 1 of VIC forced with PGF (Sheffield et al., 2006) | 0.25° | Daily | 1950–2016 | Several years | He et al. (2020) |
| *Models with satellite data assimilation* | | | | | | |
| ERA5 | ECMWF ERA5-HRES reanalysis layer 1 (0–7 cm); ASCAT soil moisture DA | 0.28° | Hourly | 1979–2020 | Several months | Hersbach et al. (2020) |
| GLEAM | GLEAM V3.3a surface layer (0–10 cm); MSWEP V2.2 $P$ forcing; ESA-CCI DA | 0.25° | Daily | 1980–2018 | 6–12 months | Martens et al. (2017) |
| HBV-ERA5+SMAPL3E | HBV forced with ERA5 $P$; SMAPL3E DA | 0.1° | 3-hourly | 2015–2020 | Several months | This study (Section 2.4) |
| HBV-IMERG+SMAPL3E | HBV forced with IMERG $P$; SMAPL3E DA | 0.1° | 3-hourly | 2015–present | Several hours | This study (Section 2.4) |
| HBV-MSWEP+SMAPL3E | HBV forced with MSWEP $P$; SMAPL3E DA | 0.1° | 3-hourly | 2015–present | Several hours[b] | This study (Section 2.4) |
| SMAPL4 | SMAP L4 V4 surface layer (0–5 cm); NASA Catchment model forced with GEOS $P$ corrected using CPC Unified (Chen et al., 2008); SMAP brightness temperature DA | 9 km | 3-hourly | 2015–present | 2–3 days | Reichle et al. (2019b); Reichle et al. (2019a) |

[a] We also evaluated versions of these products with Soil Wetness Index (SWI) filter (Wagner et al., 1999; Albergel et al., 2008) with the time lag constant $T$ set to 5 days.

[b] At a latency of hours, MSWEP does not include daily gauge corrections and is therefore of lower quality. The data evaluated here have an effective latency of several days.





**Table 2.** Median Pearson correlations ($R$) between *in situ* measurements and retrievals from ascending and descending overpasses for the single-sensor soil moisture products (Table 1). The approximate local solar time (LST) of the overpass is reported in parentheses.

| | Correlation ($R$) | |
| --- | --- | --- |
| Product | Ascending (LST) | Descending (LST) |
| AMSR2 | 0.40 (13:30) | 0.50 (01:30) |
| ASCAT | 0.41 (21:30) | 0.47 (09:30) |
| SMAPL3E | 0.65 (18:00) | 0.65 (06:00) |
| SMOS | 0.49 (06:00) | 0.48 (18:00) |

**Table A1.** The measurement networks part of the ISMN archive from which we have used *in situ* soil moisture data.

| Network | Reference(s) or website |
| --- | --- |
| ARM | www.arm.gov |
| BIEBRZA | www.igik.edu.pl |
| BNZ-LTER | Van Cleve et al. (2015) |
| COSMOS | Zreda et al. (2008, 2012) |
| CTP | Yang et al. (2013) |
| DAHRA | Tagesson et al. (2015) |
| FMI | http://fmiarc.fmi.fi |
| FR | www.inrae.fr |
| HOBE | Kang et al. (2014); Jin et al. (2014) |
| HYDROL-NET | Morbidelli et al. (2014) |
| iRON | Osenga et al. (2019) |
| LAB-net | Mattar et al. (2014) |
| MySMNet | Kang et al. (2016) |
| ORACLE | https://gisoracle.inrae.fr |
| OZNET | Smith et al. (2012) |
| REMEDHUS | http://campus.usal.es/~hidrus/ |
| RISMA | Ojo et al. (2015) |
| RSMN | http://assimo.meteoromania.ro |
| SCAN | www.wcc.nrcs.usda.gov |
| SMOSMANIA | Calvet et al. (2007); Albergel et al. (2008) |
| SNOTEL | www.wcc.nrcs.usda.gov |
| SOILSCAPE | Moghaddam et al. (2010); Moghaddam et al. (2016) |
| SWEX | Marczewski et al. (2010) |
| TERENO | Zacharias et al. (2011) |
| UDC | Loew et al. (2009) |
| USCRN | Bell et al. (2013) |
| VAS | http://nimbus.uv.es |
| WSMN | Petropoulos and McCalmont (2017) |