# Peer review of "Evaluation of 18 satellite- and model-based soil moisture products using *in situ* measurements from 826 sensors"

_Hydrology and Earth System Sciences, 2020_

## Short Comment (SC1) · 23 May 2020

The manuscript of Beck et al. evaluated the temporal dynamics of 18 state-of-the-art (quasi-)global near-surface soil moisture products. I this study very interesting and up-to-date. Overall, the paper is well organized and well written, and provides new insights about the advantages and disadvantages of different soil moisture products and on the merit of various technological and methodological innovations.

However, the introduction is not well written and more discussion and comparison to recent studies should be provided. In my opinion, the paper deserves publication once the following points are addressed with some more details.

[Figure]

line 4-12: Provide the reason why you would like to address these questions. I like your way to express your purposes of your study. However, it's not appropriate to pose so many questions here without giving any reason.

Section 2: Why these datasets are chosen out for comparison? What are main differences among the products within each group (i.e., satellites, open-loop models, and models with DA)?

The authors missed some recent publications on soil moisture evaluation. For example:

Chen, Y., & Yuan, H. (2020). Evaluation of nine sub-daily soil moisture model products over China using high-resolution in situ observations. Journal of Hydrology, 125054. https://doi.org/10.1016/j.jhydrol.2020.125054

Tavakol, A., Rahmani, V., Quiring, S. M., & Kumar, S. V. (2019). Evaluation analysis of NASA SMAP L3 and L4 and SPoRT-LIS soil moisture data in the United States. Remote Sensing of Environment, 229, 234-246. https://doi.org/10.1016/j.rse.2019.05.006

Add a review on these publications in introduction and more discussions with these papers in Section 4 will add much value to this manuscript.

line 30: What are the sensor types? Are there all FDR sensors?

Add a map showing the observation length and the frequency of in-situ observation.

Table 1: Add one column to describe the vertical layers for the soil moisture products. Since soil moisture data of model products or satellites are not representative at 5 cm, have you done some vertical interpolation?

---

## Referee Comment (RC1) · Anonymous Referee #1 · 7 Jun 2020

This paper describes the performance of various gridded soil moisture products with in situ surface soil moisture measurements. There is a lot thrown into this comparison and the methodology seems solid, but I am not sure about what we learned in the end.

1. Why is this particular subset of products selected? It is mixing spatial (horizontal & vertical) resolutions, operational and research products, etc., and makes a fair comparison questionable. Furthermore, it is not possible to stratify the results based on this random mix of features. Please provide more justification for the evaluation setup or refocus the paper.

For example, why was SMAPL3E included and not the coarser-scale L2/3 product,

why SMOS v650 and not SMOS-IC, why GLEAM, etc. There is no reason given for the chosen products, even though the various products serve different purposes and have very different characteristics (e.g. SMOS retrievals offer both SM and VOD, DA products offer much more than only surface soil moisture, just to name a few).

Perhaps this paper should move its focus towards evaluating the new MeMo product and its underlying HBV modeling system, rather than shuffling that product into a general analysis that tries to vaguely address a list of too general questions for a non-representative or inconsistent subset of data products?

Random example: what is the relative performance of the single-sensor satellite products? If "all" available soil moisture products would be compared, or some meaningful features would be targeted, then we could learn something from this, but for the 4 discussed products, more than half of the answer was already given in earlier papers and the added value of the answer in this paper is minimal.

2. The 5-day filter is used to reduce noise, but has also been used to derive root-zone soil moisture in the past. Why are the results compared to surface soil moisture and not root-zone in situ measurements? Would that not be fairer?

3. In general, there is very little mentioning of the vertical representativity of the various products. It cannot possibly be that all products produce a consistent ∼5 cm surface product. For example, how deep is the HBV soil moisture store? Is it comparable in volume to the volume observed by satellite data or other model-satellite surface soil moisture products? Due to their different wavelengths, the AMSR2, ASCAT and SMOS/SMAP products must be sensitive to different vertical surface layers. Is it fair to compare them all to the same ∼5-cm surface in situ measurements?

4. The temporal resolution is also questionable: how is it possible to do a 3-hourly evaluation for all products (p.3, L.20)? Satellites only pass over every so many days.

5. Please provide more information on the quality screening of the satellite data. The

text only mentions screening for frozen conditions, but each product comes with its own flags that need to be applied. For example, it is mentioned that AMSR2 and SMOS are more vulnerable to RFI: how did you screen these data for RFI? Did you screen for dense vegetation, topographic complexity, etc?

6. The consideration of both high and low frequency signals for the calculation of R is a good idea, but why is there no evaluation of the interannual variability, using a simple state-of-the-art anomaly R?

7. Not understood: "only HBV and the Catchment model underlying SMAPL4 have been calibrated". Is it fair to say that Catchment would be "calibrated" (for soil moisture, just like HBV?) in order to hardwire a single parameter (a constant)? Wouldn't all models then ever have been 'calibrated' to chose some hardwired parameters?

---

## Referee Comment (RC2) · Christian Massari (Referee) · 10 Jun 2020

**"Evaluation of 18 satellite- and model-based soil moisture products using in situ measurements from 826 sensors" by Beck et al.**

This is my first review of the manuscript "Evaluation of 18 satellite- and model-based soil moisture products using in situ measurements from 826 sensors".

The study is very interesting and fits well with the scope of the HESS journal. It is well written and structured with relevant research questions answered in details in the results section. The literature cited is updated and figures and tables well formatted.

Despite this I have different **MAJOR** comments the authors should seriously consider:

1. 826 sensors is quite a large number for soil moisture stations and gives the impression that this evaluation is very general. However, by looking at the locations where these sensors are located the reader realizes that the majority are located over US and Europe, that is, over very data rich regions (i.e., where models tend to perform better). I think in title is much more important to highlight where the analysis is carried out rather than the number of sensors used. This give also a clearer picture of the results obtained in the study.

2. Following point 1 the results can be a bit biased towards models (also considering the type of evaluation the authors chose, see my comment 3e) and product that require use calibration (e.g., HBV runs). The product evaluation is in practice carried out exactly where in situ observations are more dense and where are more dense more calibration stations are present. This is partly highlighted by the authors but only at the end of the document while I would add more discussion about this issue.

3. The overall methodology needs to be strongly improved and detailed as many aspects are not clear and/or not well discussed and justified:

    a. The evaluation is carried by considering the temporal dynamic which is fine for the considerations done in the paper and from previous literature (see Koster et al. 2009), however, it is not clear how the evaluation at 3 hour resolution is done for satellite data with a revisit time larger than 1 day (e.g. SMAP, SMOS) and for model forced with rainfall with daily resolution. This must be clarified.

    b. The Triple Collocation (TC) is a foundation of one integration technique (i.e., the one of MeMo) and is a well known technique for the readers this manuscript point to. I am surprised that its theoretical foundation has not introduced in a more rigorous way and the assumptions made not tested. For instance, line 24 page 4 reads "with ASCAT$_{SWI}$ and HBV-MSWEP, which are independent from each other and from the passive products". First the requirements are not the independence among the products but independence of their errors as well as their mutual linearity. These assumptions might not hold even for the products chosen (Gruber et al. 2016) as here, in addition, also SWI is systematically applied to at least two products of the triplet. This can falsify the results obtained via TC. I think some additional discussion and testing of the validity of the assumption is needed. The authors can consider the application of the Quadruple collocation technique (Gruber et al. 2016) for testing this assumption which many authors of this manuscript are familiar with.

    c. Maybe this is just a technical matter and nothing major but talking about the climatology of SMAP sounds quite weird with only four four/ five years of observations. From an evaluation point of view I think it still fine, however, from a longer perspective this climatology is likely to not consider the real climate variability.

    d. Many terms and procedures are just mentioned without specifying important details. This makes the study hardly reproducible. Examples: line 17 pag. 5 "*Temperature*

*estimates were taken from ERA5, downscaled to 0.1 and bias-corrected on a monthly basis through an additive approach*". How the downscaling and the bias correction has been done exactly? "Additionally, we calculated Pearson correlation coefficients for the low- and high-frequency fluctuations of the 3-hourly time series…". Please tell what correlations of high and low fluctuations would provide in addition to classical correlation?

e. This is an important aspect: "*We did not average sites with multiple sensors to avoid potentially introducing discontinuities in the time series.*" Line 31 pag. 6. This means that if the satellite footprint of a specific product includes multiple in situ stations multiple correlations values are considered? If so, this makes the process of evaluation very random and not really under control as different products are characterized by a different spatial sampling and might include a different number of stations. Moreover, this exacerbates the problem of biased results towards model or products working well over US as many correlation values would originates from stations located in United States with an additional penalization of other locations which have already less stations. For a fair evaluation each pixel must count one correlation value. In this respect the product collocation is a crucial aspect that has not properly discussed and described in the manuscript. For example in Su et al. (2015) and Massari et al. (2017) the co-location of the satellite data and model data was determined by nearest-neighbour association and a screening step for removing ground sensors non-representative at the coarse scale was implemented. In their study, if multiple valid stations co-located in a satellite pixel were present, the station with the highest mean correlation was retained (see section 2.6 of Su et al. 2015 for further details).

f. "*We calibrated the 7 relevant parameters of HBV using in situ soil moisture measurements between 2010 and 2019 from 177 independent sensors from the International Soil Moisture Network (ISMN) archive that were not used for performance assessment (Section 2.5; Supplement Fig. S2).*" Line 20 pag. 5. How the selection of these stations was carried out? Why 177? Why such a spatial distribution? Does a different choice provide similar results? I think all these aspects need to be clarified.

g. "*T was set to 5 days for all products, as the performance did not change markedly using different values, as also reported in previous studies*". The application of the exponential filter with a constant parameter T=5 days might be not appropriate for all the satellite products as the different products have a different vertical support. Since the calibration was carried out for the model why T was not calibrated also for the satellite products?

4. *"As forcing, we used the MSWEP precipitation dataset because of its favourable performance in numerous evaluations …. The calibrated parameter set was used for all HBV runs, including those forced with ERA5 or IMERG precipitation."*

I think proceeding in this way is not fair for the cross-validation. As HBV is basically a conceptual model, its parameters tend to correct also for errors contained in the data used to force it. Indeed, it has been largely demonstrated in the scientific literature (e.g., Zeng et al., 2018) that the impact of imperfect precipitation estimates on model efficiency can be reduced to some extent through the adjustment of model parameters. In other words, If you calibrate the parameters for MSWEP rainfall, then, when you force HBV with others precipitation inputs the results might be sub-optimal. Thus for a fair evaluation different sets of parameters should be used each one referring to the specific rainfall product used to force the hydrological model.

5. MeMo integration. The study is based on similar conceptual framework presented in Kim et al. 2018 here (maximization of correlation) with the difference that in Kim et al. correlations are calculated with a benchmark while here are obtained from TC. Beside the satisfaction of the underlying assumptions related to TC which I have discussed on point 3b, Eq. 3-5 of the study of Kim et al. demonstrates that for the maximization of R when merging two products (but this holds for multiple products also), cross-correlation terms must be taken into account (it is also demonstrated in Gruber et al. 2017 already cited in the manuscript) thus the framework described in MeMo integration is not theoretically optimal. However, if the products are

independent the framework collapses into a simple weighing average as cross-correlation are zero. I assume the authors consider null cross correlations within SMAP, SMOS and AMSR2 which I think is statistically not demonstrated. So i strongly suggest to provide some additional details and justifications about the integration framework used. This can explain why MeMo " *MeMo performed only marginally better in terms of R than the best-performing single-sensor product SMAPL3ESWI*" (Line 15 pag. 12).

6. *"The satellite products provided the least reliable soil moisture estimates and exhibited the largest regional performance differences on average, whereas the models with satellite data assimilation provided the most reliable soil moisture estimates and exhibited the smallest regional performance differences on average.".* I think the authors should highlight again here that this result is expected given the high density gauge observations used in the study area. Highlighting this is very important as for instance ground validation conducted in data-rich areas does not adequately reflect the added values of satellite observations (Dong et al. 2019).

**Minor comments:**

Line 24 pag. 3. Every satellite product contains proper quality flags for removing these low quality data while doing this with an external dataset might not guarantee optimal results. Please at least discuss this.

Line 12 pag. 4. "Three-hourly soil moisture time series of AMSR2SWI, SMAPL3ESWI, SMOSSWI". No clear how these time series are created or extracted from products having revisit times larger than 1 day. This is unknown in the paper.

Line 20 pag. 6. So the triplet is the same as above except for the presence of SMAPL3E in place of SMAPL3ESWI?

Figure 1 caption: Stations in Europe are not really visibile (e.g., Denmark). Can you make a bit darker?

Figure 2 caption: Please explain better panels b, c and d.

Line 6-9 pag. 11. I think this is the main reason.

Line 24 pag. 12, *"First, ESA-CCISWI incorporates ASCAT, which performed less well in the present evaluation, whereas".* This cannot be a reason if the integration is "optimal" as the different parent products are weighed according to their relative performance. So the second one is more likely the reason. Please rephrase or justify with more solid arguments.

Line 3 pag. 13. "*and satellite-based GPCP V1.3 Daily Analysis (Huffman et al., 2001)*" How a daily rainfall can provide 3-hourly estimates?

Line 20 pag. 14. Please explain what is the meaning of *efficiency* here.

Line 11 pag. 16. Check this sentence, it appears out of place.

Table 3: Latency of the products. Change to a more precise value or remove. Several does not provide enough information. I think ERA5 is now available with a delay of three days.

Table 3: Spatial and temporal resolution. With such a diverse range of products I suggest to replace "temporal resolution and spatial resolution" with spatial and temporal sampling.

**References**

Gruber, A., Su, C. H., Crow, W. T., Zwieback, S., Dorigo, W. A., & Wagner, W. (2016). Estimating error cross-correlations in soil moisture data sets using extended collocation analysis. Journal of Geophysical Research: Atmospheres, 121(3), 1208-1219.

*Zeng, Q., Chen, H., Xu, C. Y., Jie, M. X., Chen, J., Guo, S. L., & Liu, J. (2018). The effect of rain gauge density and distribution on runoff simulation using a lumped hydrological modelling approach. Journal of hydrology, 563, 106-122.*

*Su, C. H., Narsey, S. Y., Gruber, A., Xaver, A., Chung, D., Ryu, D., & Wagner, W. (2015). Evaluation of post-retrieval de-noising of active and passive microwave satellite soil moisture. Remote Sensing of Environment, 163, 127-139.*

*Massari, C., Su, C. H., Brocca, L., Sang, Y. F., Ciabatta, L., Ryu, D., & Wagner, W. (2017). Near real time de-noising of satellite-based soil moisture retrievals: An intercomparison among three different techniques. Remote Sensing of Environment, 198, 17-29.*

*Dong, J., Crow, W., Reichle, R., Liu, Q., Lei, F., & Cosh, M. H. (2019). A Global Assessment of Added Value in the SMAP Level 4 Soil Moisture Product Relative to Its Baseline Land Surface Model. Geophysical Research Letters, 46(12), 6604-6613.*

---

## Referee Comment (RC3) · Hylke E. Beck et al. · 16 Jun 2020

This is a very interesting and promising paper certainly useful to document the biblio-
graphical effort on soil moisture evaluation. I feel however that authors have skimmed
over some essential explanations and was sometimes wondering if I had the latest ver-
sion of the manuscript from HESSD (?). The bullet points format of the manuscript does
not help and a lot of discussion is missing prior it can be considered for publication. I
recommend major revisions, please see below an attempt to help.

Although very important, this kind of evaluation is by design almost never in favour of
the satellite based products. It has been highlighted several time in the literature in
the past decade that in data rich areas where models are highly constrained by high

quality observations, their soil moisture is of better quality that the one retrieved from spatial remote sensing. As the in situ measurements sensors you are using are largely located in those data rich areas, this should be emphasize in the manuscript.

Page 1, Lines 9-8 : a) It gives the false impressions that data assimilation brings an improvement going from 0.69 to 0.72 while models with data assimilation do not all have open-loop counterparts (the opposite being true as well). I know it is the abstract but perhaps you should already give scores that can highlight the added value of data assimilation by considering the mean R values of their open-loop counterpart (HBV+ERA5, HBV+IMERG, HBV+MSWEP). b) I am personally not a big fan of such statement in an abstract and I am not sure it is well supported by your results particularly regarding the large distribution of your scores (boxplots of figures 2 & 3) and the lack of discussions on score difference significance

Page 1, Line 14 (also Line 16 and true for many part of the manuscript): Are those differences significant? why didn't you provide confidence intervals? Also according to figure 2 it is ESA-CCI_SWI that has a median R value of 0.67 while ESA-CCI has a median R value of 0.56, please clarify. The notion of with/without SWI does not appear in the abstract (?).

Page 2, Line 14 "Additionally, many had a regional (sub-continental) focus [. . .]" I would not say yours is different (?) Particularly looking at figure 1, please clarify. Also you could add a lot of recent references that had looked at very similar dataset to like to yours. You are only slightly discussing towards the end of your manuscript, please revised

Page 2, Lines 25-26 "Furthermore, several new or recently reprocessed products have not been thoroughly evaluated yet, such as ERA5 (Hersbach et al., 2020), ERA5-Land (C3S, 2019), and ESA-CCI V04.4 (Dorigo et al., 2017)." For ERA5, Li et al have used 842 qualified sites covering 25 networks (rather recent paper I must admit): https://rmets.onlinelibrary.wiley.com/doi/10.1002/joc.6549

[Figure]

For ESA-CCI, Did you check the product website and documentation? https://www.esa-soilmoisture-cci.org/validation

Page 3, Line 1 " [. . .] from 826 sensors located primarily in the USA and Europe [. . .]" Thus as for previous studies you have mentioned the extent to which your findings can be generalized is unclear (?), please revise as this sentence could be misleading.

Page 3, Line 5 Question on SWI appears only here, seems a bit out of the blue (?) please introduce SWI earlier not to confuse readers.

Page 3, Line 14, section 2.1 I am wondering here if I have the correct version of the manuscript as several dataset are not presented? It is a general comment that you have to justify why you have used those 18 dataset and not others, otherwise it looks like cherry-picking. While some are state-of-the arts, others are self-made, please revised the choice and presentation of the dataset.

Page 3, Line 24 I assume you have used soil temperature of the first layer of soil between 1-7cm, is so please say it. Alternatively you could have discarded in situ measurements of soil moisture when associated measurements of soil temperature (if available) was < 4 dC

Page 4, Lines 16-17 Add references I appropriate

Page 5, Line 10 "The model was run twice for 2010–2019 [. . .]" Please clarify if this was done for each forcing dataset (I assume so) Page 5, Line 20 "We calibrated the 7 relevant parameters of HBV [. . .]" This will have to be discuss further already if it impacts your results wrt to the land surface model based product?

Page 6, section 2.5 Are they all using the same measurement methodology?

Page 7, figure 1 In such study this kind of global maps tend to show areas with no data more than areas with data. It is not obvious than 2 two zooms over North America and Europe add anything, perhaps you could have one figure with 3 panels, North America, Europe and Australia (?)

Also I suspect here that most of the stations in the "cold" class over North America are from the SNOTEL network located in mountainous area where the retrieval of soil moisture from space is rather complex. This should be emphasise in the text at it is biasing your results.

Page 9, figure 2 I may have missed a point but I did not understand how did you obtain 3-hourly data for e.g. ASCAT, SMOS, ESA-CCI, SMAP...please revise.

It would have been easier to have them close to one another (SWI and not SWI) on your figures but has you have several questions to answer it was probably not easy to pick up the correct order of products for those figures.

Page 11, section 3.2 My personal opinion is that this is a low pass filter smoothing the time-series, nothing more

Page 11, section 3.3 Are you R values significant? I may have missed something here but from your figures 2 and 3 (boxplots distribution) it is difficult for me to give a clear answer to this question (while you are doing it in the abstract)

Page 12, section 3.4 Line 21 "[. . .] the central Rocky Mountains [. . .]" This are usually area where it is difficult to retrieve soil moisture form space. Memo perhaps does better than ESA-CCI but is it good? are we talking about R values going from 0.2 to 0.3 or from 0.6 to 0.8? From figure 4 it is difficult to see anything (at least to me). Again, are the differences significant? Lines 22-23 Confidence interval would help Line 29 Please clarify "[. . .] from the best sensor each day[. . .]"

Page 12, Lines 31-32 Is it surprising to find the 3 calibrated HBV models leading this ranking? Again I would not claim such a best to worst ranking without discussing the significance of scores.

Page 13, figure 4 (also true for figure 5) Not sure this figure is very helpful as hardly visible (?) Perhaps you could use scatterplots, e.g. x-axis R for ESA-CCI vs in situ, y-axis R for MeMo vs in situ and then use color-codes for any classification you like.

Page 13, Line 1 ERA5 is a coupled land atmosphere system where ASCAT has been assimilated. Could you comment on the impact it may (or may not) have when using it to force HTESSEL land surface model in ERA5-Land? is it fully independent from ASCAT?

P.14, Line 1 Data intensive models could also be calibrated don't you think? I personally thing it is wrong to oppose land surface model and calibrated hydrological models. their objectives are different.

P.14, Line 8 There is more to say from such figure as figure 5 (?) e.g. discuss the geographical patterns

P.14, Lines 21-23 Please discuss if it is likely to be because of the inputs quality (AS-CAT/SMOS) or a methodological matter.

P.14, Lines 26 There is also a study showing that the assimilation of ESA CCI in GLEAM leads to a decrease of quality (Brecht et al., 2018 GMD?)

Page 14, Lines 32-33 Which was expected right?

P.15, section 3.7 Perhaps this could be moved few sections above?

P.16, Lines 16-19 In agreement with many previous studies (e.g. Albergel et al., 2010, HESS, Dorigo et a;., 2017, RSE...)

P.17, section 3.9 Perhaps worth referencing / discussing Reichle et al,. 2019 ? Verification of the SMAP Level-4 Soil Moisture Analysis Using Rainfall Observations in Australia, https://ieeexplore.ieee.org/document/8898398

---

## Author Comment (AC1) · 7 Jul 2020

The manuscript of Beck et al. evaluated the temporal dynamics of 18 state-of-the-art (quasi-)global near-surface soil moisture products. I this study very interesting and up-to-date. Overall, the paper is well organized and well written, and provides new insights about the advantages and disadvantages of different soil moisture products and on the merit of various technological and methodological innovations.

We thank Dr. Nelson for reviewing our manuscript and providing thoughtful comments.

However, the introduction is not well written and more discussion and comparison to recent studies should be provided. In my opinion, the paper deserves publication once the following points are addressed with some more details.

We appreciate the comment; we have re-read the introduction with this in mind and made some improvements.

line 4-12: Provide the reason why you would like to address these questions. I like your way to express your purposes of your study. However, it's not appropriate to pose so many questions here without giving any reason.

We agree and have added that these questions are *"frequently faced by researchers and end-users alike."* References and further background on each question is provided in the subsections discussing addressing the questions (Sections 3.1 to 3.9).

Section 2: Why these datasets are chosen out for comparison? What are main differ-ences among the products within each group (i.e., satellites, open-loop models, and models with DA)?

Good question. We have added the following to justify our product selection: "*We evaluated six products per category, which was sufficient to compare the performance among and within product categories and address the questions posed in the introduction. We only considered widely used products with (quasi-)global coverage and we attempted to keep the selection of products in each category as diverse as possible. For example, we considered products based on several major satellite missions used for global soil moisture mapping (AMSR2, ASCAT, SMAP, and SMOS), models of various type and complexity (with and without calibration), different sources of precipitation data (satellites, reanalyses, gauges, and combinations thereof), and various data merging and assimilation techniques (with different inputs).*"

The authors missed some recent publications on soil moisture evaluation. For example:

Chen, Y., & Yuan, H. (2020). Evaluation of nine sub-daily soil moisture model products over China using high-resolution in situ observations. Journal of Hydrology, 125054.https://doi.org/10.1016/j.jhydrol.2020.125054

Tavakol, A., Rahmani, V., Quiring, S. M., & Kumar, S. V. (2019). Evaluation analysis of NASA SMAP L3 and L4 and SPoRT-LIS soil moisture data in the United States. Remote Sensing of Environment, 229, 234-246. https://doi.org/10.1016/j.rse.2019.05.006

Add a review on these publications in introduction and more discussions with these papers in Section 4 will add much value to this manuscript.

*Thanks for pointing us to these very interesting studies. We have added them to the introduction and to other relevant sections of the paper. Even though our paper has already well over 200 references, the body of literature on soil moisture estimation is so vast that it's easy to miss studies.*

line 30: What are the sensor types? Are there all FDR sensors?

*We have added the following text:* *"The measurements were performed using various types of sensors, including time-domain reflectometry sensors, frequency-domain reflectometry sensors, capacitance sensors, and cosmic-ray neutron sensors, among others."*

Add a map showing the observation length and the frequency of in-situ observation.

*Please see Supplementary material Fig. S1 for a figure showing the observation length and the frequency of in situ observation.*

Table 1: Add one column to describe the vertical layers for the soil moisture products. Since soil moisture data of model products or satellites are not representative at 5 cm, have you done some vertical interpolation?

*The depths of the soil layers of the models are provided in the "Details" column. The penetration depth of microwave signals can differ significantly depending on the observation frequency and the land surface conditions, and therefore cannot be listed in the table. To improve the vertical representation of the satellite products, we used the SWI filter (see Section 2.1). We have added the following text to the revised manuscript*

to discuss the vertical support of the models: *"The vertical support is physically consistent with in situ soil moisture measurements at 5-cm depth for most models. The average depth of the soil layer (i.e., half the depth of the lower boundary) is 2.5 cm for SMAPL4, 3.5 cm for ERA5 and ERA5-Land, 5 cm for GLEAM, 8.5 cm for HBV-ERA5, 6.6 cm for HBV-IMERG, 7.3 cm for HBV-MSWEP, and 15 cm for VIC-PGF (Table 1; Supplement Table S1). The soil layers of HBV may seem too deep, especially since they represent conceptual "buckets" that can be fully filled with water, in contrast to the soil layers of the other models which additionally consist of mineral and organic matter. However, the soil layer depths of HBV were calibrated (see Section 2.3) and are thus empirically consistent with in situ measurements at 5-cm depth."*

---

## Author Comment (AC2) · 7 Jul 2020

This paper describes the performance of various gridded soil moisture products within situ surface soil moisture measurements. There is a lot thrown into this comparison and the methodology seems solid, but I am not sure about what we learned in the end.

We thank the reviewer for their thorough assessment and helpful comments.

Briefly summarized, we evaluated the largest and most diverse selection of soil moisture products to date, to the best of our knowledge. This allowed us to gain several novel insights into the relative advantages and disadvantages of a broad range of methodologies and data sources used to estimate soil moisture, as well as into techniques to evaluate the estimates. Of course, one outcome of our comparison is quantitative information on the relative performance of different products, which will be helpful for researchers deciding which product(s) to use in their analysis. However, there are several other findings:

- a smoothing filter helps to avoid disadvantaging noisy satellite products in product evaluations;

- the new ERA5 reanalysis precipitation data provide performance close to gauge-based precipitation estimates;

- satellite products perform worse in cold climates than in warmer climates, and so do model products;

- a simple, calibrated model can outperform substantially more complex, data-intensive models;

- precipitation data quality is the main factor determining the benefit of data assimilation;

- satellite data assimilation provides greater performance improvements for models with a poor soil moisture simulation efficiency;

- model calibration can be more beneficial than satellite data assimilation into an uncalibrated model;

- satellite products tend to exhibit larger regional performance differences than models;

We believe that these findings, which are succinctly summarized in the conclusions, are of value to the general readership of HESS.

1. Why is this particular subset of products selected? It is mixing spatial (horizontal & vertical) resolutions, operational and research products, etc., and makes a fair comparison questionable. Furthermore, it is not possible to stratify the results based on this random mix of features. Please provide more justification for the evaluation setup or refocus the paper.

This is a good question. We have added the following to the revised manuscript to justify our product selection: "*We evaluated six products per category, which was sufficient to compare the performance among and within product categories and address the questions posed in the introduction. We only considered widely used products with (quasi-)global coverage, and we attempted to keep the selection of products in each category as diverse as possible. For example, we considered products based on several major satellite missions used for global soil moisture mapping (AMSR2, ASCAT, SMAP, and SMOS), models of various type and complexity (with and without calibration), different sources of precipitation data (satellites, reanalyses, gauges, and combinations thereof), and various data merging and assimilation techniques (with different inputs).*"

For example, why was SMAPL3E included and not the coarser-scale L2/3 product, why SMOS v650 and not SMOS-IC, why GLEAM, etc. There is no reason given for the chosen products, even though the various products serve different purposes and have very different characteristics (e.g. SMOS retrievals offer both SM and VOD, DA products offer much more than only surface soil moisture, just to name a few).

Please see our preceding response. There are too many candidate soil moisture products available for us to include all of them in our analysis. Some admittedly subjective selection is therefore necessary.

We appreciate that different products have different design objectives and characteristics and offer different auxiliary data, but do not see how that invalidates our evaluation.

Perhaps this paper should move its focus towards evaluating the new MeMo product and its underlying HBV modeling system, rather than shuffling that product into a general analysis that tries to vaguely address a list of too general questions for anon-representative or inconsistent subset of data products?

Thanks for the comment. A paper just about MeMo and HBV would be of interest to a much smaller part of the community than the present evaluation. The MeMo product

was included primarily to assess the effectiveness of a different merging approach compared to ESA-CCI. The HBV model products were added to (i) examine how well a simple calibrated model performs, (ii) assess the impact of different precipitation forcing datasets on the overall performance, and (iii) quantify the benefits of satellite data assimilation for different precipitation forcing datasets.

We believe the nine questions posed in the paper are pertinent to numerous researchers and end users of soil moisture data and not "too general." We are not entirely sure why the reviewer refers to our way of addressing the questions "vague" as we provide clear, concise, and well-referenced objectives and findings.

As an aside, HBV is not underlying the MeMo product, they are completely independent products.

Random example: what is the relative performance of the single-sensor satellite products? If "all" available soil moisture products would be compared, or some meaningful features would be targeted, then we could learn something from this, but for the 4 discussed products, more than half of the answer was already given in earlier papers and the added value of the answer in this paper is minimal.

This example pertains to just one of the nine questions addressed in the paper. Earlier papers did not apply smoothing filters, stratified the results in different ways, most did not explicitly assess high- and low-frequency fluctuations, and most did not compare the performance of these single-sensor products to other types of products. As such, we believe our analysis does provide significant added value. Furthermore, even if part of our answer to this question was already given in earlier papers, we do not see it as a bad thing to replicate the findings of previous papers.

2. The 5-day filter is used to reduce noise, but has also been used to derive root-zone soil moisture in the past. Why are the results compared to surface soil moisture and not root-zone in situ measurements? Would that not be fairer?

We actually applied the 5-day filter to make the comparison with the *in situ* measurements at 5-cm depth more fair. The 5-day filter serves two purposes, to reduce noise and to deepen the vertical support of the superficial satellite observations (not to the root zone but to approximately 5 cm). Without the 5-day filter, the satellite products would perform, on average, significantly worse than the two other major product categories, and some particularly noisy satellite products would be severely disadvantaged (e.g., SMOS; see Fig. 2). If we had used *in situ* measurements of the

root zone as reference we probably would have used a filter with a longer temporal window.

3. In general, there is very little mentioning of the vertical representativity of the various products. It cannot possibly be that all products produce a consistent~5 cm surface product. For example, how deep is the HBV soil moisture store? Is it comparable in volume to the volume observed by satellite data or other model-satellite surface soil moisture products? Due to their different wavelengths, the AMSR2, ASCAT and SMOS/SMAP products must be sensitive to different vertical surface layers. Is it fair to compare them all to the same ~5-cm surface in situ measurements?

Thanks for the comment. We believe our study represents a fair comparison. The 5-day filter deepens the vertical support to make the superficial satellite observations more representative of *in situ* measurements at 5-cm depth (please see our previous response). Previous soil moisture product evaluations tended to compare soil moisture retrievals directly to *in situ* measurements at 5-cm depth, and therefore may have underestimated the 'true' skill of products. We considered optimizing the time lag constant *T* for each product but decided against this, because we wanted to make statements about the accuracy of the original data, not a post-processed product.

We have added the following text to the revised manuscript regarding the vertical support of the models: *"The vertical support is physically consistent with in situ soil moisture measurements at 5-cm depth for most models. The average depth of the soil layer (i.e., half the depth of the lower boundary) is 2.5 cm for SMAPL4, 3.5 cm for ERA5 and ERA5-Land, 5 cm for GLEAM, 8.5 cm for HBV-ERA5, 6.6 cm for HBV-IMERG, 7.3 cm for HBV-MSWEP, and 15 cm for VIC-PGF (Table 1; Supplement Table S1). The soil layers of HBV may seem too deep, especially since they represent conceptual "buckets" that can be fully filled with water, in contrast to the soil layers of the other models which additionally consist of mineral and organic matter. However, the soil layer depths of HBV were calibrated (see Section 2.3) and are thus empirically consistent with in situ measurements at 5-cm depth."*

4. The temporal resolution is also questionable: how is it possible to do a 3-hourly evaluation for all products (p.3, L.20)? Satellites only pass over every so many days.

Thank you for the comment. We have added the following text into the revised manuscript: *"For the satellite products without SWI filter, we matched the instantaneous soil moisture retrievals with coincident 3-hourly in situ measurements to compute the R values."*

5. Please provide more information on the quality screening of the satellite data. The text only mentions screening for frozen conditions, but each product comes with its own flags that need to be applied. For example, it is mentioned that AMSR2 and SMOS are more vulnerable to RFI: how did you screen these data for RFI? Did you screen for dense vegetation, topographic complexity, etc?

We appreciate the suggestion. We will provide more information about the quality flags used for the satellite products in the revised manuscript. Thanks for the suggestion.

6. The consideration of both high and low frequency signals for the calculation of R is a good idea, but why is there no evaluation of the interannual variability, using a simple state-of-the-art anomaly R?

The "state-of-the-art anomaly R" measures both the (seasonal-scale) interannual variability and short-term deviations from the long-term mean seasonal cycle.  The skill of short-term variations in the soil moisture products is assessed in our high-frequency filter.

We did not separately evaluate the (seasonal-scale) interannual variability due to the short temporal span of some of the products (less than 5 years), which precludes us from calculating reliable correlation coefficients.

7. Not understood: "only HBV and the Catchment model underlying SMAPL4 have been calibrated". Is it fair to say that Catchment would be "calibrated" (for soil moisture,just like HBV?) in order to hardwire a single parameter (a constant)? Wouldn't all models then ever have been 'calibrated' to chose some hardwired parameters?

Both HBV and Catchment have been explicitly calibrated  against independent *in situ* soil moisture measurements by optimizing a certain performance metric. The same may be true but has not been similarly documented for the other models included in the evaluation. The calibration procedure of HBV is described in Section 2.3, while the calibration procedure of Catchment is described in Reichle et al. (2019b).

---

## Author Comment (AC3) · 7 Jul 2020

This is my first review of the manuscript "Evaluation of 18 satellite- and model-based soil moisture products using in situ measurements from 826 sensors".The study is very interesting and fits well with the scope of the HESS journal. It is well written and structured with relevant research questions answered in details in the results section. The literature cited is updated and figures and tables well formatted.

We thank Dr. Massari for his thorough assessment of our manuscript.

Despite this I have different MAJOR comments the authors should seriously consider:

1. 826 sensors is quite a large number for soil moisture stations and gives the impression that this evaluation is very general. However, by looking at the locations where these sensors are located the reader realizes that the majority are located over US and Europe, that is, over very data rich regions (i.e., where models tend to perform better). I think in title is much more important to highlight where the analysis is carried out rather than the number of sensors used. This give also a clearer picture of the results obtained in the study.

Thanks for the suggestion. We considered replacing *"from 826 sensors"* with *"from the US, Europe, and Australia"* in the title. However, since this would make the title less concise, we did not make this change. We do, however, clearly highlight in the paper that our results may not generalize to the entire global land surface and have devoted an entire subsection (3.9) to this issue.

We do not fully agree with the generalization that models perform better over data-rich regions, as this depends on the precipitation forcing used to drive the models. Our evaluation includes six models with non-gauge-based precipitation forcings (ERA5, ERA5-Land, HBV-ERA5 with and without data assimilation, and HBV-IMERG with and without data assimilation), and the performance of these models is largely representative of data-poor regions.

2. Following point 1 the results can be a bit biased towards models (also considering the type of evaluation the authors chose, see my comment 3e) and product that require use calibration (e.g., HBV runs). The product evaluation is in practice carried out exactly where in situ observations are more dense and where are more dense more calibration stations are present. This is partly highlighted by the authors but only at the end of the document while I would add more discussion about this issue.

Thanks for the comment. We have changed several existing sentences and added the following sentence to Section 3.9: *"The calibrated models (HBV and the Catchment model underlying SMAPL4) may, however, perform slightly worse in regions with climatic and physiographic conditions dissimilar to the in situ sensors used for calibration (but probably still better than the uncalibrated models)."*

3.The overall methodology needs to be strongly improved and detailed as many aspects are not clear and/or not well discussed and justified:

a.The evaluation is carried by considering the temporal dynamic which is fine for the considerations done in the paper and from previous literature (see Koster et al. 2009), however, it is not clear how the evaluation at 3 hour resolution is done for satellite data with a revisit time larger than 1 day (e.g. SMAP, SMOS) and for model forced with rainfall with daily resolution. This must be clarified.

We agree and have added the following text to explain this more clearly in the revised manuscript: *"For the satellite products without SWI filter, we matched the instantaneous soil moisture retrievals with coincident 3-hourly in situ measurements to compute the R values."*

b.The Triple Collocation (TC) is a foundation of one integration technique (i.e., the one of MeMo) and is a well known technique for the readers this manuscript point to. I am surprised that its theoretical foundation has not introduced in a more rigorous way and the assumptions made not tested. For instance, line 24 page 4 reads "with ASCATSWI and HBV-MSWEP, which are independent from each other and from the passive products". First the requirements are not the independence among the products but independence of their errors as well as their mutual linearity. These assumptions might not hold even for the products chosen (Gruber et al. 2016) as here, in addition, also SWI is systematically applied to at least two products of the triplet. This can falsify the results obtained via TC. I think some additional discussion and testing of the validity of the assumption is needed. The authors can consider the application of the Quadruple collocation technique (Gruber et al. 2016) for testing this assumption which many authors of this manuscript are familiar with.

We thank the reviewer for his comment. We do not entirely agree with the statement that *"the requirements are not the independence among the products but independence of their errors,"* because if the products are fully independent, it follows that the errors will be fully independent as well. Unless of course the reference is imperfect (which is

the case if *in situ* data are used as reference), in which case the errors reflect both the product and the reference.

We recognize that the error independence assumption and other assumptions may not be fully satisfied in our study and we have therefore added the following statement to the revised paper: "*Triple collocation-based merging techniques rely on several assumptions (linearity, stationarity, error orthogonality, and zero cross-correlation; Gruber et al., 2016) which are generally difficult to fully satisfy in practice, affecting the optimality of the merging procedure.*"

We carefully examined the Quadruple Collocation (QC) methodology presented in Gruber et al. (2016). They note that QC still requires *"zero error cross covariance between some specific data set combinations"* (Section 2.4), which means that expert judgement is still needed to determine which products have correlated errors and which don't prior to estimating the correlations between two products. Pan et al. (2015) also highlighted the need for expert pre-judgement. QC is therefore only useful to estimate the correlation after already having "assumed" that particular products are more likely to be correlated than others. In light of this, we believe QC offers limited independent insight into the TC assumptions. The developer of QC makes a similar statement in Gruber et al. (2017): *"Recently, Gruber et al. (2016) proposed an extension to TCA where the inclusion of more than three data sets in the analysis allows for — at least partly — resolving nonzero error cross-correlation structures, yet a demonstration of the robustness of the method on a global scale is still pending. Therefore, one may for practical reasons neglect error cross correlations between different active or passive data sets at the cost of non optimal SNR improvements, or make a conservative educated guess for error cross-correlation levels for data sets where they are expected.*"

The application of the SWI filter was necessary to temporally match the different satellite products, which would not have been possible using instantaneous retrievals at non-overlapping irregular times (Gruber et al., 2020). We agree, however, that the SWI filter does not need to be applied to both satellite products in the triplets, and therefore in the revised manuscript we use unfiltered ASCAT data.

c.Maybe this is just a technical matter and nothing major but talking about the climatology of SMAP sounds quite weird with only four four/ five years of observations. From an evaluation point of view I think it still fine, however, from a longer perspective this climatology is likely to not consider the real climate variability.

We agree and have replaced *"climatologies"* with *"averages."*

d.Many terms and procedures are just mentioned without specifying important details. This makes the study hardly reproducible. Examples: line 17 pag. 5 "Temperature estimates were taken from ERA5, downscaled to 0.1 and bias-corrected on a monthly basis through an additive approach". How the downscaling and the bias correction has been done exactly? "Additionally, we calculated Pearson correlation coefficients for the low- and high-frequency fluctuations of the 3-hourly time series...". Please tell what correlations of high and low fluctuations would provide in addition to classical correlation?

Thanks for bringing this up; we read the manuscript again to make sure no details are missing. We added the following to explain the ERA5 correction: *"To improve the representation of mountainous regions and ameliorate potential biases, the ERA5 air temperature data were matched on a monthly climatological basis using an additive (as opposed to multiplicative) approach to the comprehensive station-based WorldClim climatology (V2; 1-km resolution; Fick and Hijmans, 2017)."*

The following sentence was added to better highlight the added value of the Pearson correlation coefficients for the low- and high-frequency fluctuations: *"Additionally, to quantify the performance of the products at different time scales, we calculated Pearson correlation coefficients for the low-frequency fluctuations (i.e., the slow variability at monthly and longer time scales; $R_{lo}$) and the high-frequency fluctuations (i.e., the fast variability at 3-hourly to monthly time scales; $R_{hi}$)."*

e.This is an important aspect: "We did not average sites with multiple sensors to avoid potentially introducing discontinuities in the time series." Line 31 pag. 6. This means that if the satellite footprint of a specific product includes multiple in situ stations multiple correlations values are considered? If so, this makes the process of evaluation very random and not really under control as different products are characterized by a different spatial sampling and might include a different number of stations. Moreover, this exacerbates the problem of biased results towards model or products working well over US as many correlation values would originates from stations located in United States with an additional penalization of other locations which have already less stations. For a fair evaluation each pixel must count one correlation value. In this respect the product collocation is a crucial aspect that has not properly discussed and described in the manuscript. For example in Su et al. (2015) and Massari et al. (2017) the co-location of the satellite data and model data was determined by nearest-neighbour association and a screening step for removing ground sensors

non-representative at the coarse scale was implemented. In their study, if multiple valid stations co-located in a satellite pixel were present, the station with the highest mean correlation was retained (see section 2.6 of Su et al. 2015 for further details).

We thank the reviewer for this thoughtful comment. This issue is commonly referred to as the collocation issue (Gruber et al., 2020) and unfortunately there are no satisfactory solutions, particularly when the products have such a wide range of grid-cell and footprint sizes. After much deliberation we decided not to change the current approach for the following reasons:

1. A coarser spatial sampling should, in our opinion, be penalized (as is currently the case), since it reflects a technical limitation in the ability of the product to represent heterogeneous areas.

2. We believe that grid-cells or footprints with multiple *in situ* sensors should be assigned more weight (as is currently the case), because the presence of multiple sensors reduces the sampling uncertainty and thus leads to a more reliable performance estimate.

3. The removal of *in situ* sensors that are not representative of the coarse scale is not straightforward in our evaluation due to the substantial variety in model grid-cell and satellite footprint sizes. We are not in favor of resampling all products to a common grid as this would penalize products with a higher spatial resolution.

4. The removal of 'unrepresentative' *in situ* sensors is further confounded by the fact that the location of satellite footprints varies over time (i.e., the footprint of today's satellite overpass is not exactly the same as the footprint of the next overpass). Su et al. (2015) and Massari et al. (2017) did not have this issue as their products were all gridded.

5. Retaining only the *in situ* sensors with the best performance may paint an overly rosy picture of the products.

We would like to note that our approach has also been used by numerous other researchers (e.g., Albergel et al., 2012; Karthikayan et al., 2017; Al-Yaari et al., 2019), which thus implicitly agreed with our view. Nevertheless, we agree about the importance of highlighting that several dense measurement networks exert a strong influence on the overall results and we therefore expanded the first sentence of Section 3.9 as follows: *"The large majority (98 %) of the in situ soil moisture measurements used as*

*reference in the current study were from dense monitoring networks in the USA and Europe (Fig. 1) and therefore our results will be most applicable to these regions.*"

f."We calibrated the 7 relevant parameters of HBV using in situ soil moisture measurements between 2010 and 2019 from 177 independent sensors from the International Soil Moisture Network (ISMN) archive that were not used for performance assessment (Section 2.5; Supplement Fig. S2)." Line 20 pag. 5. How the selection of these stations was carried out? Why 177? Why such a spatial distribution? Does a different choice provide similar results? I think all these aspects need to be clarified.

We have added the following to the revised manuscript: *"These sensors did not have enough measurements during the evaluation period (March 31, 2015, to September 16, 2019) and thus were available for an independent calibration exercise."* A different selection of *in situ* sensors would have provided similar results due to the low degrees of freedom (just 7 parameters were calibrated using 177 sensors). Note that HBV has been recalibrated for ERA5 and IMERG in the revised paper.

g."T was set to 5 days for all products, as the performance did not change markedly using different values, as also reported in previous studies". The application of the exponential filter with a constant parameter T=5 days might be not appropriate for all the satellite products as the different products have a different vertical support. Since the calibration was carried out for the model why T was not calibrated also for the satellite products?

We strongly considered optimizing the time lag constant *T* for each product in the revised manuscript but in the end decided against this for two main reasons. First, we did not want to deviate too much from the original data because we want to make statements about the accuracy of the original data, not a post-processed product. Secondly, we did not want to give the satellite products an unfair advantage compared to the uncalibrated models, which would likely also benefit from the application of the SWI filter (though likely not as much).

The calibration of HBV was carried out because the model cannot be run without calibration, as it is a conceptual model with parameters that do not represent physical properties of the land surface. Note that we added the following regarding the generalization of the performance of the calibrated models to Section 3.9: *"The calibrated models (HBV and the Catchment model underlying SMAPL4) may, however, perform slightly worse in regions with climatic and physiographic conditions dissimilar to*

*the in situ sensors used for calibration (but likely still better than the uncalibrated models)."*

4."As forcing, we used the MSWEP precipitation dataset because of its favourable performance in numerous evaluations .... The calibrated parameter set was used for all HBV runs, including those forced with ERA5 or IMERG precipitation." I think proceeding in this way is not fair for the cross-validation. As HBV is basically a conceptual model, its parameters tend to correct also for errors contained in the data used to force it. Indeed, it has been largely demonstrated in the scientific literature (e.g., Zeng et al., 2018) that the impact of imperfect precipitation estimates on model efficiency can be reduced to some extent through the adjustment of model parameters. In other words, If you calibrate the parameters for MSWEP rainfall, then, when you force HBV with others precipitation inputs the results might be sub-optimal. Thus for a fair evaluation different sets of parameters should be used each one referring to the specific rainfall product used to force the hydrological model.

Our initial reason for not recalibrating HBV for ERA5 and IMERG was that we did not expect the resulting parameters to realistically represent the transformation of precipitation to soil moisture, because ERA5 and IMERG do not incorporate any gauge data and exhibit systematic errors (in mean, occurrence, and magnitude; Beck et al., 2019a). Conversely, the calibration of MSWEP has likely resulted in parameters that relatively realistically represent the transformation of precipitation to soil moisture, since MSWEP incorporates vast amounts of daily gauge data and exhibits almost no systematic errors in the study area (Beck et al., 2019a).

However, since we agree that the recalibration of HBV for ERA5 and IMERG might potentially lead to a small performance improvement, we followed the reviewer's suggestion and carried out the recalibration. The following text was added: *"To avoid giving one of the precipitation datasets an unfair advantage, we recalibrated the model for each of the three precipitation datasets (ERA5, IMERG, and MSWEP)."* The negligible performance improvement after calibration for ERA5 and IMERG (0.00 and 0.01, respectively) probably reflects the low degrees of freedom (just 7 model parameters were calibrated using data from 177 sensors) and thus limited ability of the parameters to correct for systematic errors.

5.MeMo integration. The study is based on similar conceptual framework presented in Kim et al. 2018 here (maximization of correlation) with the difference that in Kim et al. correlations are calculated with a benchmark while here are obtained from TC. Beside the satisfaction of the underlying assumptions related to TC which I have discussed on

point 3b, Eq. 3-5 of the study of Kim et al. demonstrates that for the maximization of R when merging two products (but this holds for multiple products also), cross-correlation terms must be taken into account (it is also demonstrated in Gruber et al. 2017 already cited in the manuscript) thus the framework described in MeMo integration is not theoretically optimal. However, if the products are independent the framework collapses into a simple weighing average as cross-correlation are zero. I assume the authors consider null cross correlations within SMAP, SMOS and AMSR2 which I think is statistically not demonstrated. So i strongly suggest to provide some additional details and justifications about the integration framework used. This can explain why MeMo " MeMo performed only marginally better in terms of R than the best-performing single-sensor product SMAPL3ESWI" (Line 15 pag. 12).

We do indeed, implicitly, assume null cross-correlations among AMSR2, SMAPL3E, and SMOS. This is an assumption to all TC applications that may not be fully met, similar to the assumption of perfectly Gaussian distributions. The null cross-correlations assumption cannot be formally tested as the truth is not known. One could evaluate the correlation in deviations versus *in situ* data but of course they do not represent the truth either and they are not available everywhere, so this does not solve the issue.

6. "The satellite products provided the least reliable soil moisture estimates and exhibited the largest regional performance differences on average, whereas the models with satellite data assimilation provided the most reliable soil moisture estimates and exhibited the smallest regional performance differences on average.". I think the authors should highlight again here that this result is expected given the high density gauge observations used in the study area. Highlighting this is very important as for instance ground validation conducted in data-rich areas does not adequately reflect the added values of satellite observations (Dong et al. 2019).

Thanks for the comment. Even when excluding the three models with data assimilation using gauge-corrected precipitation forcings (GLEAM, SMAPL4, HBV-MSWEP+SMAPL3E), the remaining three models with data assimilation (ERA5, HBV-ERA5+SMAPL3E, and HBV-IMERG+SMAPL3E) still provide more reliable soil moisture estimates and smaller regional performance differences on average. This conclusion is thus not simply attributable to the inclusion of gauge observations in some of the precipitation forcings.

Minor comments:

Line 24 pag. 3. Every satellite product contains proper quality flags for removing these low quality data while doing this with an external dataset might not guarantee optimal results. Please at least discuss this.

We will expand our discussion of this.

Line 12 pag. 4. "Three-hourly soil moisture time series of AMSR2SWI, SMAPL3ESWI, SMOSSWI". No clear how these time series are created or extracted from products having revisit times larger than 1 day. This is unknown in the paper.

The last paragraph of Section 2.1 explains that the SWI filter was applied on a 3-hourly basis and that *"the SWI at time t was only calculated if ≥1 retrievals were available in the interval (t−T; t] and ≥3 retrievals were available in the interval [t−3T; t−T]."* Application of the SWI filter is thus certainly possible for products with revisit times longer than 1 day.

Line 20 pag. 6. So the triplet is the same as above except for the presence of SMAPL3E in place of SMAPL3ESWI?

This is correct.

Figure 1 caption: Stations in Europe are not really visibile (e.g., Denmark). Can you make a bit darker?

Thank you for the comment. We have increased the size of the stations and completely revised the figures.

Figure 2 caption: Please explain better panels b, c and d.

We have expanded the caption with a few additional details.

Line 6-9 pag. 11. I think this is the main reason.

The vertical representativeness could well be the main reason, however, we believe the noise reduction is also an important reason, given the often substantial seemingly random variability between consecutive instantaneous retrievals.

Line 24 pag. 12, "First, ESA-CCISWI incorporates ASCAT, which performed less well in the present evaluation, whereas". This cannot be a reason if the integration is "optimal"

as the different parent products are weighed according to their relative performance. So the second one is more likely the reason. Please rephrase or justify with more solid arguments.

*The reviewer is right in theory; as discussed earlier in our response, given the difficulty of satisfying all triple collocation assumptions, our merging approach is unlikely to be fully "optimal," and we did not claim it was. For this reason, the inclusion of a product of lower quality results in a performance degradation. As mentioned before, we have added the following statement to the preceding paragraph to highlight this: "Triple collocation-based merging techniques rely on several assumptions (linearity, stationarity, error orthogonality, and zero cross-correlation; Gruber et al., 2016) which are generally difficult to fully satisfy in practice, affecting the optimality of the merging procedure."*

Line 3 pag. 13. "and satellite-based GPCP V1.3 Daily Analysis (Huffman et al., 2001)" How a daily rainfall can provide 3-hourly estimates?

*Good question. This is explained in Section 2.1: "Since the evaluation was performed at a 3-hourly resolution, we downscaled the two products with a daily temporal resolution (VIC-PGF and GLEAM) to a 3-hourly resolution using nearest neighbor resampling." We realize that this is not ideal, but there was no other solution.*

Line 20 pag. 14. Please explain what is the meaning of efficiency here.

*Thanks for the comment. By efficiency we refer to how realistically the model represents the transformation of precipitation into soil moisture. We have rephrased "the model efficiency" to "the soil moisture simulation efficiency."*

Line 11 pag. 16. Check this sentence, it appears out of place.

*Deleted, thanks.*

Table 3: Latency of the products. Change to a more precise value or remove. Several does not provide enough information. I think ERA5 is now available with a delay of three days.

*We have provided more precise latency values. The latency of ERA5 appears to be 6 days at this moment.*

Table 3: Spatial and temporal resolution. With such a diverse range of products I suggest to replace "temporal resolution and spatial resolution" with spatial and temporal sampling.

Done.

References

Gruber, A., Su, C. H., Crow, W. T., Zwieback, S., Dorigo, W. A., & Wagner, W. (2016). Estimating error cross-correlations in soil moisture data sets using extended collocation analysis. Journal of Geophysical Research: Atmospheres, 121(3), 1208-1219.

Zeng, Q., Chen, H., Xu, C. Y., Jie, M. X., Chen, J., Guo, S. L., & Liu, J. (2018). The effect of rain gauge density and distribution on runoff simulation using a lumped hydrological modelling approach. Journal of hydrology, 563, 106-122.

Su, C. H., Narsey, S. Y., Gruber, A., Xaver, A., Chung, D., Ryu, D., & Wagner, W. (2015). Evaluation of post-retrieval de-noising of active and passive microwave satellite soil moisture. Remote Sensing of Environment, 163, 127-139.

Massari, C., Su, C. H., Brocca, L., Sang, Y. F., Ciabatta, L., Ryu, D., & Wagner, W. (2017). Near real time de-noising of satellite-based soil moisture retrievals: An intercomparison among three different techniques. Remote Sensing of Environment, 198, 17-29.

Dong, J., Crow, W., Reichle, R., Liu, Q., Lei, F., & Cosh, M. H. (2019). A Global Assessment of Added Value in the SMAP Level 4 Soil Moisture Product Relative to Its Baseline Land Surface Model. Geophysical Research Letters, 46(12), 6604-6613.

---

## Author Comment (AC4) · 7 Jul 2020

This is a very interesting and promising paper certainly useful to document the biblio-graphical effort on soil moisture evaluation. I feel however that authors have skimmed over some essential explanations and was sometimes wondering if I had the latest version of the manuscript from HESSD (?). The bullet points format of the manuscript does not help and a lot of discussion is missing prior it can be considered for publication. I recommend major revisions, please see below an attempt to help.

We thank the reviewer for their thorough assessment and helpful comments.

Although very important, this kind of evaluation is by design almost never in favour of the satellite based products. It has been highlighted several time in the literature in the past decade that in data rich areas where models are highly constrained by high quality observations, their soil moisture is of better quality that the one retrieved from spatial remote sensing. As the in situ measurements sensors you are using are largely located in those data rich areas, this should be emphasize in the manuscript.

We agree that this issue affects many previous studies. We designed our study to give the satellite products a fair opportunity in two ways:

1. We included six models with non-gauge-based precipitation forcings (ERA5, ERA5-Land, HBV-ERA5 with and without data assimilation, and HBV-IMERG with and without data assimilation). The performance of these models is largely representative of data-poor areas.

2. We evaluated versions of the satellite products processed with SWI filter which generally performed substantially better (Section 3.2). Previous soil moisture product evaluations tended to compare instantaneous soil moisture retrievals directly to the *in situ* measurements, and may therefore have underestimated the 'true' skill of satellite products.

However, despite this, the satellite products still generally performed worse.

We agree with the reviewer that it is important to highlight that models with gauge-based precipitation forcings may not perform as well in data-poor areas, which we have done multiple times in the paper:

- *"It should be kept in mind, however, that these studies, including the present one, used in situ soil moisture measurements from regions with dense rain gauge networks, and hence likely overestimate model performance (Dong et al., 2019)."*

- *"In sparsely gauged areas the four models using precipitation forcings that incorporate daily gauge observations (GLEAM, HBV-MSWEP, HBV-MSWEP+SMAPL3E, and SMAPL4; Table 1) will inevitably exhibit lower performance (but not necessarily lower than the other models)."*

Page 1, Lines 9-8 : a) It gives the false impressions that data assimilation brings an improvement going from 0.69 to 0.72 while models with data assimilation do not all have open-loop counterparts (the opposite being true as well). I know it is the abstract but perhaps you should already give scores that can highlight the added value of data assimilation by considering the mean R values of their open-loop counterpart (HBV+ERA5, HBV+IMERG, HBV+MSWEP). b) I am personally not a big fan of such statement in an abstract and I am not sure it is well supported by your results particu-larly regarding the large distribution of your scores (boxplots of figures 2 & 3) and the lack of discussions on score difference significance

We appreciate the comment. Since the abstract is already quite long we won't be able to present median $R$ improvement scores for each of the products with and without data assimilation. As suggested, we have deleted the statement referred to by the reviewer.

We have added probability ($p$) values (calculated using the Kruskal-Wallis test) in the manuscript to Table 1 where we compare the performance of ascending and descending overpasses of the single-sensor products. However, we follow the soil moisture product validation recommendations set out by Gruber et al. (2020) and avoid making any statement or interpretation about statistical significance or non-significance, because *"a label of statistical significance does not mean or imply that an association or effect is highly probable, real, true, or important. Nor does a label of statistical nonsignificance lead to the association or effect being improbable, absent, false, or unimportant."*

We did not present $p$-values for all 254 product combinations for all three performance metrics ($R$, $R_{hi}$, and $R_{lo}$) and did not explicitly report $p$-values when comparing the medians scores of different products, as this would significantly hamper the readability of the paper. Additionally, we carried out some experiments using the Kruskal-Wallis test on synthetic $R$ distributions with properties similar to the actual $R$ distributions, and found that even small differences in median $R$ of just 0.02 tend to be statistically significant at the $p$=0.05 level, whereas greater differences of 0.03 tend to be statistically significant at the $p$=0.001 level. Thus, the reviewer can safely assume that differences in median $R$ of ≥0.02 will be statistically significant at (at least) the $p$=0.05 level.

Regarding the medians of the major product categories (discussed in section 3.8), these are all significantly different at at least the $p=10^{-11}$ level.

Page 1, Line 14 (also Line 16 and true for many part of the manuscript): Are those differences significant? why didn't you provide confidence intervals? Also according to figure 2 it is ESA-CCI_SWI that has a median R value of 0.67 while ESA-CCI has a median R value of 0.56, please clarify. The notion of with/without SWI does not appear in the abstract (?).

The SWI subscript was indeed missing in the abstract and was added to ESA-CCI and the other satellite products in the revised manuscript. Additionally, we added a line about the SWI results. Thanks for the comment.

The difference in median $R$ between ESA-CCI$_{SWI}$ and MeMo is quite large and indeed highly statistically significant ($p=10^{-7}$). As explained in the preceding response, we prefer to refrain from making statements about statistical significance or non-significance.

Page 2, Line 14 "Additionally, many had a regional (sub-continental) focus [ . . .]" I would not say yours is different (?) Particularly looking at figure 1, please clarify. Also you could add a lot of recent references that had looked at very similar dataset to like to yours. You are only slightly discussing towards the end of your manuscript, please revised

There are numerous soil moisture product evaluations that focused on a single country or a small area (i.e., a sub-continental region), whereas we tried to use all available *in situ* data globally to draw the most generalizable conclusions possible. Our *in situ* data covers the entire conterminous US and thus can be considered at least "continental." We recognize, of course, that the coverage of the *in situ* sensors is far from fully global, and to we have devoted an entire subsection to discussing the generalizability of our results. Our study has already well over 200 references and we cite numerous recent studies that also use ISMN data as reference. We are not sure which recent references are missing.

Page 2, Lines 25-26 "Furthermore, several new or recently reprocessed products have not been thoroughly evaluated yet, such as ERA5 (Hersbach et al., 2020), ERA5-Land (C3S, 2019), and ESA-CCI V04.4 (Dorigo et al., 2017)." For ERA5, Li et al have used 842 qualified sites covering 25 networks (rather recent paper I must admit):

https://rmets.onlinelibrary.wiley.com/doi/10.1002/joc.6549 For ESA-CCI, Did you check the product website and documentation? https://www.esa-soilmoisture-cci.org/validation

Thanks for pointing us to the paper and website which are both very interesting. However, our paper was already finalized before Li et al. (2020) appeared online. We were aware of the online ESA-CCI evaluation, but we did not include it in the paper primarily because it has not been peer-reviewed.

Page 3, Line 1 "[. . .] from 826 sensors located primarily in the USA and Europe [. . .]" Thus as for previous studies you have mentioned the extent to which your findings can be generalized is unclear (?), please revise as this sentence could be misleading.

Agreed. We have replaced *"and thus the extent to which their findings can be generalized is unclear"* with *"potentially leading to conclusions with limited generalizability."*

Page 3, Line 5 Question on SWI appears only here, seems a bit out of the blue (?)please introduce SWI earlier not to confuse readers.

We agree, and have added the following to introduce the SWI: *"There is also still uncertainty around [...] the impact of smoothing filters such as the Soil Wetness index (SWI; Wagner et al., 1999; Albergel et al., 2008) on the performance ranking of products."*

Page 3, Line 14, section 2.1 I am wondering here if I have the correct version of the manuscript as several dataset are not presented? It is a general comment that you have to justify why you have used those 18 dataset and not others, otherwise it looks like cherry-picking. While some are state-of-the arts, others are self-made, please revised the choice and presentation of the dataset.

All products are introduced in Table 1, which we refer to in the first sentence of Section 2.1. We have added the following to justify our selection of products: *"We evaluated six products per category, which was sufficient to compare the performance among and within product categories and address the questions posed in the introduction. We only considered widely used products with (quasi-)global coverage and we attempted to keep the selection of products in each category as diverse as possible. For example, we considered products based on several major satellite missions used for global soil moisture mapping (AMSR2, ASCAT, SMAP, and SMOS), models of various type and complexity (with and without calibration), different sources of precipitation data*

*(satellites, reanalyses, gauges, and combinations thereof), and various data merging and assimilation techniques (with different inputs)."*

Page 3, Line 24 I assume you have used soil temperature of the first layer of soil between 1-7cm, is so please say it. Alternatively you could have discarded in situ measurements of soil moisture when associated measurements of soil temperature (if available) was < 4 dC

We agree and have added 0–7 cm in reference to the ERA5 soil temperature estimates.

Page 4, Lines 16-17 Add references I appropriate

We would be happy to add relevant references but we are not aware of any. This is a relatively simple part of our methodology that we believe can be understood and replicated without references.

Page 5, Line 10 "The model was run twice for 2010–2019 [...]" Please clarify if this was done for each forcing dataset (I assume so)

Yes, the initialization was performed for each precipitation dataset. In the revised manuscript HBV is recalibrated for each precipitation dataset. We have added the following text: *"To avoid giving one of the precipitation datasets an unfair advantage, we recalibrated the model for each of the three precipitation datasets (ERA5, IMERG, and MSWEP)."*

Page 5, Line 20 "We calibrated the 7 relevant parameters of HBV [...]" This will have to be discuss further already if it impacts your results wrt to the land surface model based product?

We have added the following to the revised manuscript: *"The calibrated models (HBV and the Catchment model underlying SMAPL4) may, however, perform slightly worse in regions with climatic and physiographic conditions dissimilar to the in situ sensors used for calibration (but likely still better than the uncalibrated models)."* Section 3.7 discusses the benefits and limitations of model calibration in detail, including implications with respect to the land surface model-based products, as suggested by the reviewer.

Page 6, section 2.5 Are they all using the same measurement methodology?

Thanks for the comment. We added the following text: *"The measurements were performed using various types of sensors, including time-domain reflectometry sensors, frequency-domain reflectometry sensors, capacitance sensors, and cosmic-ray neutron sensors, among others."*

Page 7, figure 1 In such study this kind of global maps tend to show areas with no data more than areas with data. It is not obvious than 2 two zooms over North America and Europe add anything, perhaps you could have one figure with 3 panels, North America,Europe and Australia (?)

Agreed; we have revised this figure (as well as the other figures) as proposed by the reviewer, but with four panels instead of three (Alaska, Europe, conterminous US, and Southeastern Australia). Thanks for the suggestion.

Also I suspect here that most of the stations in the "cold" class over North America are from the SNOTEL network located in mountainous area where the retrieval of soil moisture from space is rather complex. This should be emphasise in the text at it is biasing your results.

We agree; thanks for the comment. The retrieval may indeed be more complex in cold regions, which we mention in the paper: *"the confounding influence of dense vegetation cover (de Rosnay et al., 2006; Gruhier et al., 2008; Dorigo et al., 2010), highly organic soils (Zhang et al., 2019b), and standing water (Ye et al., 2015; Du et al., 2018) on soil moisture retrievals."* The influence of mountainous terrain on the retrievals is also mentioned in the paper: *"Most satellite products performed worse in terms of R in areas of steep terrain (Fig. 2d), consistent with previous evaluations (Paulik et al.,2014; Karthikeyan et al., 2017a; Ma et al., 2019), and attributed to the confounding effects of relief on the upwelling microwave brightness temperature observed by the radiometer (Mialon et al., 2008; Pulvirenti et al., 2011; Guo et al., 2011)."*

An additional explanation for the lower performance in cold regions (missing from our original submission) may be that the sensors are less representative of the coarse scale of the products. We therefore added the following: *"it could also be that the in situ measurements are [...] less representative of satellite footprints or model grid-cells."*

Page 9, figure 2 I may have missed a point but I did not understand how did you obtain 3-hourly data for e.g. ASCAT, SMOS, ESA-CCI, SMAP...please revise.

*This was indeed not clearly explained. We have added the following text: "For the satellite products without SWI filter, we matched the instantaneous soil moisture retrievals with coincident 3-hourly in situ measurements to compute the R values."*

It would have been easier to have them close to one another (SWI and not SWI) on your figures but has you have several questions to answer it was probably not easy to pick up the correct order of products for those figures.

*We agree that having the SWI and non-SWI products close to each other in the figure would be useful for answering the SWI-related question but less useful for the other questions addressed in the study.*

Page 11, section 3.2 My personal opinion is that this is a low pass filter smoothing the time-series, nothing more

*We agree with this observation; the SWI filter is in essence a low-pass filter smoothing the time series.*

Page 11, section 3.3 Are you R values significant? I may have missed something here but from your figures 2 and 3 (boxplots distribution) it is difficult for me to give a clear answer to this question (while you are doing it in the abstract)

*The large majority of R values are highly statistically significant, since an R value of just 0.14 tends to be needed to obtain a statistically significant correlation (at the 0.05 level) for a sample size of 200 (the minimum sample size before an R value is calculated in this study; see Figure 1). Our R values are, however, generally much higher (Fig. 2) and our sample sizes much greater, and therefore our R values will be much more statistically significant.*

[Figure]

**Figure 1.** Plot showing the minimum value of Pearson's correlation coefficient ($R$) that would be significant at the 0.05 level for a given sample size. Source: https://commons.wikimedia.org/wiki/File:Correlation_significance.svg.

As explained in the beginning of this response letter, following recommendations of Gruber et al. (2020) we refrain from making statements about statistical significance or non-significance.

Page 12, section 3.4 Line 21 "[. . .] the central Rocky Mountains [. . .]" This are usually area where it is difficult to retrieve soil moisture form space. Memo perhaps does better than ESA-CCI but is it good? are we talking about R values going from 0.2 to 0.3 orfrom 0.6 to 0.8? From figure 4 it is difficult to see anything (at least to me). Again, are the differences significant? Lines 22-23 Confidence interval would help Line 29 Please clarify "[. . .] from the best sensor each day[. . .]"

Please see Fig. 2d of our paper for median $R$ values for mountainous versus flat areas (denoted by the letters S and F, respectively) for the different products. The median $R$ is 0.61 for ESA-CCI$_{SWI}$ versus 0.73 for MeMo, which is a substantial difference (statistically significant at the $p=10^{-6}$ level).

Page 12, Lines 31-32 Is it surprising to find the 3 calibrated HBV models leading this ranking? Again I would not claim such a best to worst ranking without discussing the significance of scores.

This was somewhat surprising given the simplicity of HBV and the fact that HBV has been designed for runoff estimation in cold regions. Conversely, numerous studies have demonstrated the flexibility and effectiveness of HBV, and the model has been calibrated against *in situ* soil moisture measurements. See our discussion in the second paragraph of the subsection in question: *"This demonstrates that soil moisture estimates from complex, data-intensive models (H-TESSEL underlying ERA5 and ERA5-Land, GLEAM, and the Catchment model underlying SMAPL4) are not necessarily more accurate than those from relatively simple, calibrated models (HBV)."* Note that we have devoted an entire subsection to discussing the benefits and limitations of calibration (Section 3.7).

Page 13, figure 4 (also true for figure 5) Not sure this figure is very helpful as hardly visible (?) Perhaps you could use scatterplots, e.g. x-axis R for ESA-CCI vs in situ,y-axis R for MeMo vs in situ and then use color-codes for any classification you like.

We appreciate the comment but a scatterplot would not tell us *where* the products perform better or worse and thus would be much less informative. We could indeed use color codes to denote the locations, but we feel a map is more clear. That said, we have completely redesigned the figure and hope it is more useful now.

Page 13, Line 1 ERA5 is a coupled land atmosphere system where ASCAT has been assimilated. Could you comment on the impact it may (or may not) have when using it to force HTESSEL land surface model in ERA5-Land? is it fully independent from ASCAT?

The assimilation of ASCAT soil moisture is unlikely to have influenced the precipitation generated by ERA5, given (i) the small influence of the assimilation on the soil moisture simulations (Muñoz Sabater et al., 2019) and (ii) the vast amounts of other observations (ground and satellite) also assimilated (Hersbach et al., 2020).

P.14, Line 1 Data intensive models could also be calibrated don't you think? I personally thing it is wrong to oppose land surface model and calibrated hydrological models. Their objectives are different.

We may be misunderstanding the comment, but we do not consider it unfair to include both land surface models and calibrated hydrological models in the same evaluation. For users simply looking for the most accurate product — probably the most common type of end-user — the data source or modeling approach is not important. We fully agree that the design objectives can be different and that data-intensive models can be calibrated as well. The calibration of computationally demanding models is however more challenging, as mentioned at the end of Section 3.7.

P.14, Line 8 There is more to say from such figure as figure 5 (?) e.g. discuss the geographical patterns

Thank you for the suggestion. We have added the following: "*For HBV-IMERG, the greatest improvements were found over the central Rocky Mountains (Fig. 5), where IMERG performs relatively poorly (Beck et al., 2019a).*"

P.14, Lines 21-23 Please discuss if it is likely to be because of the inputs quality (AS-CAT/SMOS) or a methodological matter.

We explain in the sentence thereafter that it is probably a methodological issue: *"They attributed this to the adverse impact of simultaneously assimilated screen-level temperature and relative humidity observations on the soil moisture estimates."*

P.14, Lines 26 There is also a study showing that the assimilation of ESA CCI inGLEAM leads to a decrease of quality (Brecht et al., 2018 GMD?)

We suspect the reviewer might be referring to Martens et al. (2016). However, this study shows small (not negligible) improvements in the soil moisture simulations after DA.

Page 14, Lines 32-33 Which was expected right?

This was indeed in accordance with our expectations, but this has not been explicitly discussed in previous studies (to our knowledge).

P.15, section 3.7 Perhaps this could be moved few sections above?

We appreciate the comment. However, since we compare the benefits of model calibration and data assimilation in this section, we have to discuss the data assimilation results first. It is therefore not possible to move this subsection.

P.16, Lines 16-19 In agreement with many previous studies (e.g. Albergel et al., 2010,HESS, Dorigo et a;., 2017, RSE...)

We agree and list eight previous studies that agree with our results: *"Our performance ranking of the major product categories is consistent with previous studies for the conterminous USA (Liu et al., 2011; Kumar et al., 2014; Fang et al., 2016; Dong et al., 2020), Europe (Naz et al., 2019), and the globe (Albergel et al., 2012; Tian et al., 2019; Dong et al., 2019)."*

P.17, section 3.9 Perhaps worth referencing / discussing Reichle et al,. 2019 ? Verification of the SMAP Level-4 Soil Moisture Analysis Using Rainfall Observations in Australia, https://ieeexplore.ieee.org/document/8898398

Thanks for the suggestion. We are not sure which statement of Section 3.9 is supported by the results of Reichle et al. (2019). Note that the author of that study, Rolf Reichle, is also co-author of the present study.

---

## Author Comment (AC5) · 7 Jul 2020

Dear Prof. Dr. Hendricks-Franssen,

We would like to sincerely thank you for handling the manuscript and the reviewers and Korbin Nelson for their valuable comments. We have responded to the comments online. We have already prepared a revised version of the m/s, which we hope to resubmit. The main changes are as follows:

1. Explained the calculation of the correlations using instantaneous satellite soil moisture retrievals; in response to all reviewers.

2. Added Kruskal-Wallis test probability values to Table 1; in response to reviewer 3.

3. Added a justification for our product selection; in response to reviewers 1 and 3 and Korbin Nelson.

4. Added information about the sensor types included in the ISMN archive; in response to reviewer 3 and Korbin Nelson.

5. Recalibrated HBV for ERA5 and IMERG and modified the discussion; in response to reviewers 2 and 3.

6. Discussed the layer depths of the models in the main text; in response to reviewer 1 and Korbin Nelson.

7. Added a statement about the generalizability of the performance of the calibrated models; in response to reviewers 2 and 3.

8. Added more precise latency values to Table 1; in response to reviewer 2.

9. Improved all maps (Figs. 1, 4, 5, and S2); in response to reviewer 3.

10. Provided information about the product quality flags; in response to reviewer 1.

11. Added a statement regarding the difficulty of satisfying the assumptions underlying triple collocation techniques; in response to reviewer 2.

12. Added details about the ERA5 temperature bias correction and the low- and high-frequency correlation coefficients; in response to reviewer 2.

13. Added details about the *in situ* data used for the calibration of HBV; in response to reviewer 2.

Sincerely,

Hylke Beck (on behalf of all co-authors)

---

## Referee Report (RR1)

**Review #2 rebuttal**

This is my second review of the manuscript "Evaluation of 18 satellite- and model-based soil moisture products using in situ measurements from 826 sensors". Some of the issues that made the manuscript not really clear at the beginning have been clarified and the manuscript has been improved in this respect, however, there are still some MAJOR and MODERATE pending issues the authors should address.

In the following:

  a) my new comments and replies to authors are written in bold red
  b) authors replies to previous comments are in *black italic*
  c) Old comments given by myself are in **bold black**

My comments are listed below:

1) MAJOR

   This comment is related to the clarification about the 3-hourly Pearson correlation coefficient.

   I am a bit surprised the authors decide to use 3-hourly sampling to compute temporal correlation given that 13 out of the 18 products have temporal resolution >= 1 day (the majority of the models have native resolution equal to 3 hours given that they are forced by 3-hours rainfall, however, all satellite derived products plus one model forced by GPCP and GLEAM have resolution larger or equal than one day. The problem is that this forces the authors to downscale the majority of the products to something that is far from their original resolution. Of course a 3-hourly product has its strength but this can be still highlighted in the manuscript.

   Anyway, I am still fine with this approach but I have some doubts on how the downscaling has been carried out given that no details are found in the paper. For example I report below a couple of questions:

   - assuming one product has observations every day like GLEAM, the three hourly product results from the downscaling is a product having the same daily value for all the eight 3-hourly intervals? Is this obtained product compared with the 3-hourly in situ observations then?

- for satellite observations the exponential filter seems to be used as an interpolator to bring the information of satellite passes (even every three days for products like SMOS) to 3-hourly sampling. The obtained 3-hourly products are compared with 3-hourly in situ observations?

If so, it is likely that this creates an unfair evaluation between products with temporal sampling equal to 3 hours and those having native resolution larger or equal than one day. Indeed, the interpolation (downscaling) within such long temporal windows can yield significant interpolation errors. Please provide some more information on how the downscaling has been carried out and on the impact of interpolation errors on the correlation.

**2) MODERATE**

I am still not convinced about the title. Just mentioning the number of sensors does not reflect where the validation has been carried out. However, this is my personal opinion and I leave the authors and the editor the last decision on that.

**3) MODERATE**

I give my reply to the authors below.

**Old comment 2:**

Following point 1 the results can be a bit biased towards models (also considering the type of evaluation the authors chose, see my comment 3e) and product that require use calibration (e.g., HBV runs). The product evaluation is in practice carried out exactly where in situ observations are more dense and where are more dense more calibration stations are present. This is partly highlighted by the authors but only at the end of the document while I would add more discussion about this issue.

*Reply by the authors:*

*We do not fully agree with the generalization that models perform better over data-rich regions, as this depends on the precipitation forcing used to drive the models. Our evaluation includes six models with non-gauge-based precipitation forcings (ERA5, ERA5-Land, HBV-ERA5 with and without data assimilation, and HBV-IMERG with and without data assimilation), and the performance of these models is largely representative of data-poor regions.*

Thanks for the comment. We have changed several existing sentences and added the following sentence to Section 3.9: *"The calibrated models (HBV and the*

*Catchment model underlying SMAPL4) may, however, perform slightly worse in regions with climatic and physiographic conditions dissimilar to the in situ sensors used for calibration (but probably still better than the uncalibrated models)."*

**My reply:**

- **I understand your point, however, model precipitation (ERA5 for instance) assimilates a large number of ground observations like 2-m temperature and humidity and, in US -- where most of the stations of the study are located -- also the NCEP Stage IV analysis rainfall which combines rain gauges and radars estimates (Lopez et al. 2011). Therefore, models forced by ERA5 have to be considered something not far from gauge-corrected products at least in the US and will likely to perform better with respect to what they can do within data scarce regions.**
  **Moreover, the calibration can significantly help to improve the performance of conceptual models like HBV where the soil moisture station density is high.**

- *"but probably still better than the uncalibrated models".* **Please either demonstrate this statement or provide a reference, otherwise remove.**

4) **MODERATE. I still doubt about the validation exercise. I can finally accept this approach, but I provided below some of the reasons behind my doubts.**

**Old comment 3.e**

**This is an important aspect: "We did not average sites with multiple sensors to avoid potentially introducing discontinuities in the time series." Line 31 pag. 6. This means that if the satellite footprint of a specific product includes multiple in situ stations multiple correlations values are considered? If so, this makes the process of evaluation very random and not really under control as different products are characterized by a different spatial sampling and might include a different number of stations. Moreover, this exacerbates the problem of biased results towards model or products working well over US as many correlation values would originates from stations located in United States with an additional penalization of other locations which have already less stations. For a fair evaluation each pixel must count one correlation value. In this respect the product collocation is a crucial aspect that has not properly discussed and described in the manuscript. For example in Su et al. (2015) and Massari et al. (2017) the co-location**

**of the satellite data and model data was determined by nearest-neighbour association and a screening step for removing ground sensors non-representative at the coarse scale was implemented. In their study, if multiple valid stations co-located in a satellite pixel were present, the station with the highest mean correlation was retained (see section 2.6 of Su et al. 2015 for further details).**

*Reply of the reviewers:*

*We thank the reviewer for this thoughtful comment. This issue is commonly referred to as the collocation issue (Gruber et al., 2020) and unfortunately there are no satisfactory solutions, particularly when the products have such a wide range of grid-cell and footprint sizes. After much deliberation we decided not to change the current approach for the following reasons:*

1. *A coarser spatial sampling should, in our opinion, be penalized (as is currently the case), since it reflects a technical limitation in the ability of the product to represent heterogeneous areas.*

   **My reply:**

   **I partially agree as due to temporal stability issues (Vachaud, 1984) it is likely the temporal dynamic of the stations is similar (so the results in terms of correlation are potentially less affected than any other metric like bias and error).**

   We believe that grid-cells or footprints with multiple *in situ* sensors should be assigned more weight (as is currently the case), because the presence of multiple sensors reduces the sampling uncertainty and thus leads to a more reliable performance estimate.

   **My reply:**

   **True but again this will favor calibrated runs.**

2. The removal of *in situ* sensors that are not representative of the coarse scale is not straightforward in our evaluation due to the substantial variety in model grid-cell and satellite footprint sizes. We are not in favor of resampling all products to a common grid as this would penalize products with a higher spatial resolution.

   **My reply:**

**I agree with this.**

3. The removal of 'unrepresentative' *in situ* sensors is further confounded by the fact that the location of satellite footprints varies over time (i.e., the footprint of today's satellite overpass is not exactly the same as the footprint of the next overpass). Su et al. (2015) and Massari et al. (2017) did not have this issue as their products were all gridded.

   **My reply:**

   **Fine, but this means that the stations considered at different time steps will vary in your evaluation from time to time? Can you provide more details on this? If so this must be specified.**

4. Retaining only the *in situ* sensors with the best performance may paint an overly rosy picture of the products.

   **My reply:**

   **I do not fully agree with this. For the same temporal stability issue described above, the stations with the best performance are potentially the ones more representative of the spatial mean related to the domain of the satellite footprint. So it is not unfair to consider them but, to my opinion, it would be the best thing to do.**

Vachaud, G., Passerat de Silans, A., Balabanis, P., & Vauclin, M. (1985). Temporal stability of spatially measured soil water probability density function. *Soil Science Society of America Journal*, *49*(4), 822-828.

**5) MAJOR**

**Old comment 3.**

**The overall methodology needs to be strongly improved and detailed as many aspects are not clear and/or not well discussed and justified:**

   a. **The evaluation is carried by considering the temporal dynamic which is fine for the considerations done in the paper and from previous literature (see Koster et al. 2009), however, it is not clear how the evaluation at 3 hour resolution is done for satellite data with a revisit time larger than 1 day (e.g. SMAP, SMOS) and for model forced with rainfall with daily resolution. This must be clarified.**

*Reply by the authors:*

*We agree and have added the following text to explain this more clearly in the revised manuscript: "For the satellite products without SWI filter, we matched the instantaneous soil moisture retrievals with coincident 3-hourly in situ measurements to compute the R values."*

**My reply:**

**Thanks, this is clearer now for products where the Exponential filter was not applied. However, where the Exponential filter has been applied please refer to my comment 1 above.**

6) **MODERATE/MAJOR**

    **Old Comment:**

    **"T was set to 5 days for all products, as the performance did not change markedly using different values, as also reported in previous studies". The application of the exponential filter with a constant parameter T=5 days might be not appropriate for all the satellite products as the different products have a different vertical support. Since the calibration was carried out for the model why T was not calibrated also for the satellite products?**

*We strongly considered optimizing the time lag constant T for each product in the revised manuscript but in the end decided against this for two main reasons. First, we did not want to deviate too much from the original data because we want to make statements about the accuracy of the original data, not a post-processed product. Secondly, we did not want to give the satellite products an unfair advantage compared to the uncalibrated models, which would likely also benefit from the application of the SWI filter (though likely not as much).*

**My replies:**

*"First, we did not want to deviate too much from the original data because we want to make statements about the accuracy of the original data, not a post-processed product."*
**I think that downscaling satellite time series at 3-hourly resolution (from original revisit time of more than one day) by the application of the exponential filter (it does not matter whether T is 5, 7 or 3 days) already provides a strongly post-processed product.**

*"Secondly, we did not want to give the satellite products an unfair advantage compared to the uncalibrated models, which would likely also benefit from the application of the SWI filter (though likely not as much)."*

**R: Well, HBV is calibrated with 7 parameters on the 177 stations so I do not see limitations on doing the same calibration of the exponential filter with one single parameter (which, in its original formulation, is itself a conceptual approach to obtain root zone soil moisture). That is, the 177 calibration stations could be used to calibrate the parameter *T* which best fits observations in terms of correlation.**

The calibration of HBV was carried out because the model cannot be run without calibration, as it is a conceptual model with parameters that do not represent physical properties of the land surface. Note that we added the following regarding the generalization of the performance of the calibrated models to Section 3.9: *"The calibrated models (HBV and the Catchment model underlying SMAPL4) may, however, perform slightly worse in regions with climatic and physiographic conditions dissimilar to the in situ sensors used for calibration (but likely still better than the uncalibrated models)."*

**R: remove the** *"but likely still better than the uncalibrated models"* **as it is not demonstrated or provide a reference to validate this statement.**

**7) MODERATE/MAJOR**

**Old comment 6.**

**"The satellite products provided the least reliable soil moisture estimates and exhibited the largest regional performance differences on average, whereas the models with satellite data assimilation provided the most reliable soil moisture estimates and exhibited the smallest regional performance differences on average.". I think the authors should highlight again here that this result is expected given the high density gauge observations used in the study area. Highlighting this is very important as for instance ground validation conducted in data-rich areas does not adequately reflect the added values of satellite observations (Dong et al. 2019).**

Reply by the authors:

*Thanks for the comment. Even when excluding the three models with data assimilation using gauge-corrected precipitation forcings (GLEAM, SMAPL4, HBV-MSWEP+SMAPL3E), the remaining three models with data assimilation (ERA5, HBV-ERA5+SMAPL3E, and HBV-IMERG+SMAPL3E) still provide more reliable soil moisture estimates and smaller regional performance differences on average. This conclusion is thus not simply attributable to the inclusion of gauge observations in some of the precipitation forcings.*

**My Reply:**

**All ERA5 runs contain gauge precipitation in the US where most of the stations are located, so in practice, only HBV-IMERG+SMAPL3E (which is a calibrated product) has in theory no gauge information in it.**

**8) MODERATE**

**Old comment:**

**Line 24 pag. 12, "First, ESA-CCISWI incorporates ASCAT, which performed less well in the present evaluation, whereas". This cannot be a reason if the integration is "optimal" as the different parent products are weighed according to their relative performance. So the second one is more likely the reason. Please rephrase or justify with more solid arguments.**

*Reply of the authors:*

*The reviewer is right in theory; as discussed earlier in our response, given the difficulty of satisfying all triple collocation assumptions, our merging approach is unlikely to be fully "optimal," and we did not claim it was. For this reason, the inclusion of a product of lower quality results in a performance degradation. As mentioned before, we have added the following statement to the preceding paragraph to highlight this: "Triple collocation-based merging techniques rely on several assumptions (linearity, stationarity, error orthogonality, and zero cross-correlation; Gruber et al., 2016) which are generally difficult to fully satisfy in practice, affecting the optimality of the merging procedure."*

**My reply:**

**I think that ESA-CCI contains so many products and the merging procedure so complex that it is impossible to affirm that the guilty is one product rather than another one. ESA-CCI contains also SMOS which in Figure 2 is worse/equal to ASCAT but I do not feel to say the guilty is SMOS. Please revise this sentence or provide a more solid argument to state that.**

**9) MODERATE**

**Old comment:**

**Line 3 pag. 13. "and satellite-based GPCP V1.3 Daily Analysis (Huffman et al., 2001)" How a daily rainfall can provide 3-hourly estimates?**

Answer by the authors:

Good question. This is explained in Section 2.1: *"Since the evaluation was performed at a 3-hourly resolution, we downscaled the two products with a daily temporal resolution (VIC-PGF and GLEAM) to a 3-hourly resolution using nearest neighbor resampling."* We realize that this is not ideal, but there was no other solution.

**My reply:**

**Can you clarify it better? Do you downscale GPCP daily to 3 hourly data? So the daily value is divided by 8 to have consistent daily accumulations?**

**10) MODERATE**

**I think it is important to provide some plots of the time series for instance for one/two locations (to put at least in the supplementary information) to better visualize the impact of the downscaling procedure and the visual comparison between the products.**

---

## Author Response (AR2)

**Comments to the Author:**

Dear Dr Beck,

Your manuscript "Evaluation of 18 satellite- and model-based soil moisture products using in situ measurements from 826 sensors" has been subjected now to re-review by two of the original reviewers. One reviewer recommends acceptance of the manuscript and another reviewer major revision. Please handle carefully the remaining comments. In particular, it will be important to highlight the details of the downscaling approach to get 3-hourly products. The authors should also motivate better other decisions in their work, which could have favored model-based approaches. These limitations should at least be discussed in more detail. I suggest moderate revision.

In your answer to the main points and also the many detailed comments, please indicate how comments have been handled exactly, indicating also whether text has been deleted and what the position of newly included text blocks is. Preferably, cite the newly added text in the manuscript. I am looking forward to the new version of the paper.

Best regards,

Harrie-Jan Hendricks Franssen – editor

We would like to thank you for handling the manuscript and the two reviewers for their assessments. Below we provide responses to Dr. Massari's comments in green font, to explain how we believe our evaluation is fair and that we are not favoring model-based approaches. The following main changes have been made:

1. added a clarification regarding the downscaling of VIC-PGF and GLEAM from 3-hourly to daily;

2. added a figure illustrating the downscaling approach (the SWI filter) used to get the 3-hourly satellite products; and

3. improved the discussion of MeMo versus ESA-CCI performance.

**Review #2 rebuttal**

**This is my second review of the manuscript "Evaluation of 18 satellite- and model-based soil moisture products using in situ measurements from 826 sensors". Some of the issues that made the manuscript not really clear at the beginning have been clarified and**

**the manuscript has been improved in this respect, however, there are still some MAJOR and MODERATE pending issues the authors should address.**

 **In the following:**

   a) **my new comments and replies to authors are written in bold red**

   b) **authors replies to previous comments are in** black italic

   c) **Old comments given by myself are inbold black**

**My comments are listed below:**

   1) **MAJOR**

   **This comment is related to the clarification about the 3-hourly Pearson correlation coefficient.**

   **I am a bit surprised the authors decide to use 3-hourly sampling to compute temporal correlation given that 13 out of the 18 products have temporal resolution >= 1 day (the majority of the models have native resolution equal to 3 hours given that they are forced by 3-hours rainfall, however, all satellite derived products plus one model forced by GPCP and GLEAM have resolution larger or equal than one day. The problem is that this forces the authors to downscale the majority of the products to something that is far from their original resolution. Of course a 3-hourly product has its strength but this can be still highlighted in the manuscript.**

The satellite products do not have a daily (or coarser) temporal resolution in the conventional sense, since they represent instantaneous observations rather than integrated averages. If we would perform the evaluation at a daily time scale, we would be comparing daily *in situ* averages to instantaneous observations, which would be suboptimal. Instead, we are comparing 3-hourly averages to instantaneous observations, which should be more appropriate.

The fact that we downscaled two of the products (VIC-PGF and GLEAM) from daily to 3-hourly for the evaluation did not affect the robustness of the results. This is because, unlike precipitation time series, soil moisture time series tend to exhibit strong autocorrelation, resulting in relatively small differences between 3-hourly and daily time series. Note that the authors of these two products are included as co-authors in the present study.

That said, we agree with the reviewer that if a product has a daily (or coarser) resolution that this is a weakness that should be reflected in the evaluation results, as is currently the case.

**Anyway, I am still fine with this approach but I have some doubts on how the downscaling has been carried out given that no details are found in the paper. For example I report below a couple of questions:**

Thanks for your comment.

- **assuming one product has observations every day like GLEAM, the three hourly product results from the downscaling is a product having the same daily value for all the eight 3-hourly intervals? Is this obtained product compared with the 3-hourly in situ observations then?**

Indeed; this is explained in Section 2.1: *"Since the evaluation was performed at a 3-hourly resolution, we downscaled the two products with a daily temporal resolution (VIC-PGF and GLEAM) to a 3-hourly resolution using nearest neighbor resampling."* To avoid confusion, we have added the following text: *"(resulting in replication of the daily value for all 3-hourly periods on each day)".*

- **for satellite observations the exponential filter seems to be used as an interpolator to bring the information of satellite passes (even every three days for products like SMOS) to 3-hourly sampling. The obtained 3-hourly products are compared with 3-hourly in situ observations?**

Indeed; this is explained in Section 2.1: *"To deepen the vertical support of the superficial satellite observations and suppress noise, we also evaluated 3-hourly versions of the satellite products processed using the SWI exponential smoothing filter."* Note that we have added the following figure (Fig. 1 of the revised manuscript) to illustrate the SWI filter:

[Figure]

**If so, it is likely that this creates an unfair evaluation between products with temporal sampling equal to 3 hours and those having native resolution larger or equal than one day. Indeed, the interpolation (downscaling) within such long temporal windows can yield significant interpolation errors. Please provide some more information on how the**

**downscaling has been carried out and on the impact of interpolation errors on the correlation.**

The reviewer suggests that we are giving the model products an *"unfair advantage"* compared to the satellite products because the interpolation (i.e., the SWI exponential filter) introduces errors. We respectfully disagree: the satellite products perform much better with SWI filter than without — the filter thus reduces the errors (see Figs. 3 and 4 of the revised manuscript). We devoted a subsection to discussion of the impact of the filter (see Section 3.2).

**2) MODERATE**

**I am still not convinced about the title. Just mentioning the number of sensors does not reflect where the validation has been carried out. However, this is my personal opinion and I leave the authors and the editor the last decision on that.**

We appreciate the comment. We have once again considered changing the title but we feel this title best describes the content and novelty of the study.

**3) MODERATE**

**I give my reply to the authors below.**

**Old comment 2:**

**Following point 1 the results can be a bit biased towards models (also considering the type of evaluation the authors chose, see my comment 3e) and product that require use calibration (e.g., HBV runs). The product evaluation is in practice carried out exactly where in situ observations are more dense and where are more dense more calibration stations are present. This is partly highlighted by the authors but only at the end of the document while I would add more discussion about this issue.**

*Reply by the authors:*

*We do not fully agree with the generalization that models perform better over data-rich regions, as this depends on the precipitation forcing used to drive the models. Our evaluation includes six models with non-gauge-based precipitation forcings (ERA5, ERA5-Land, HBV-ERA5 with and without data assimilation, and HBV-IMERG with and without data assimilation), and the performance of these models is largely representative of data-poor regions.*

Thanks for the comment. We have changed several existing sentences and added the following sentence to Section 3.9: *"The calibrated models (HBV and the Catchment model underlying SMAPL4) may, however, perform slightly worse in regions with climatic and physiographic*

*conditions dissimilar to the in situ sensors used for calibration (but probably still better than the uncalibrated models).”*

**My reply:**

- **I understand your point, however, model precipitation (ERA5 for instance) assimilates a large number of ground observations like 2-m temperature and humidity and, in US -- where most of the stations of the study are located -- also the NCEP Stage IV analysis rainfall which combines rain gauges and radars estimates (Lopez et al. 2011). Therefore, models forced by ERA5 have to be considered something not far from gauge-corrected products at least in the US and will likely to perform better with respect to what they can do within data scarce regions.**

We respectfully disagree that *“ERA5 [should be considered] something not far from gauge-corrected products [...] in the US”* for the following reasons:

1. the impact of the precipitation data assimilation is limited overall due to the large amount of other (ground and satellite) observations already assimilated (Lopez, 2013);

2. radar data were not used west of 105°W for quality reasons (Lopez, 2011);

3. very good performance in terms of precipitation was also found in regions without assimilated gauge observations (e.g., Nevada; Beck et al., 2019a, their Fig. 4b; Lopez, 2013, their Fig. 3); and

4. the values of actual gauge-corrected precipitation products, such as MSWEP and CHIRPS, are in complete agreement with gauges (in regions with a high gauge density), which is not the case for ERA5 due to physical constraints in terms of energy and water availability.

We note that one of the co-authors of the present study co-developed ERA5.

*Lopez, P.: Direct 4D-Var Assimilation of NCEP Stage IV Radar and Gauge Precipitation Data at ECMWF, Mon. Weather Rev., 139,2098–2116, 2011.*

*Lopez, P.: Experimental 4D-Var Assimilation of SYNOP Rain Gauge Data at ECMWF, Mon. Weather Rev., 141, 1527–1544,2013.*

**Moreover, the calibration can significantly help to improve the performance of conceptual models like HBV where the soil moisture station density is high.**

- *“but probably still better than the uncalibrated models”*. **Please either demonstrate this statement or provide a reference, otherwise remove.**

We prefer not to remove this statement. We have performed an independent evaluation of the calibrated model (see Sections 2.3 and 3.7) demonstrating that the performance also translates to completely independent soil moisture probes, which already have different climatic and physiographic conditions. We consider it highly unlikely that the benefit of the calibration suddenly disappears at other (slightly more different) locations. The statement is thus, in our opinion, accurate.

The reviewer's statement that *"the calibration can significantly help to improve the performance of conceptual models like HBV"* is problematic as it implies that calibration is an optional step for conceptual models. It is not as these models tend to have parameters without clear physical interpretation.

**4) MODERATE. I still doubt about the validation exercise. I can finally accept this approach, but I provided below some of the reasons behind my doubts.**

**Old comment 3.e**

**This is an important aspect: "We did not average sites with multiple sensors to avoid potentially introducing discontinuities in the time series." Line 31 pag. 6. This means that if the satellite footprint of a specific product includes multiple in situ stations multiple correlations values are considered? If so, this makes the process of evaluation very random and not really under control as different products are characterized by a different spatial sampling and might include a different number of stations. Moreover, this exacerbates the problem of biased results towards model or products working well over US as many correlation values would originates from stations located in United States with an additional penalization of other locations which have already less stations. For a fair evaluation each pixel must count one correlation value. In this respect the product collocation is a crucial aspect that has not properly discussed and described in the manuscript. For example in Su et al. (2015) and Massari et al. (2017) the co-location**

**of the satellite data and model data was determined by nearest-neighbour association and a screening step for removing ground sensors non-representative at the coarse scale was implemented. In their study, if multiple valid stations co-located in a satellite pixel were present, the station with the highest mean correlation was retained (see section 2.6 of Su et al. 2015 for further details).**

*Reply of the reviewers:*

*We thank the reviewer for this thoughtful comment. This issue is commonly referred to as the collocation issue (Gruber et al., 2020) and unfortunately there are no satisfactory solutions,*

*particularly when the products have such a wide range of grid-cell and footprint sizes. After much deliberation we decided not to change the current approach for the following reasons:*

1.  *A coarser spatial sampling should, in our opinion, be penalized (as is currently the case), since it reflects a technical limitation in the ability of the product to represent heterogeneous areas.*

**My reply:**

**I partially agree as due to temporal stability issues (Vachaud, 1984) it is likely the temporal dynamic of the stations is similar (so the results in terms of correlation are potentially less affected than any other metric like bias and error).**

Thank you for the comment.

We believe that grid-cells or footprints with multiple *in situ* sensors should be assigned more weight (as is currently the case), because the presence of multiple sensors reduces the sampling uncertainty and thus leads to a more reliable performance estimate.

**My reply:**

**True but again this will favor calibrated runs.**

We appreciate the comment but we do not fully agree that this will favor the products based on calibrated models. The calibration freedom for HBV is so low (just 7 parameters were calibrated) that it does not matter much (if at all) which *in situ* sensors we used. Furthermore, many *in situ* sensors used for calibration were not regions with dense monitoring networks (see the supplementary information).

2.  The removal of *in situ* sensors that are not representative of the coarse scale is not straightforward in our evaluation due to the substantial variety in model grid-cell and satellite footprint sizes. We are not in favor of resampling all products to a common grid as this would penalize products with a higher spatial resolution.

**My reply:**

**I agree with this.**

We are happy to hear this.

3.  The removal of 'unrepresentative' *in situ* sensors is further confounded by the fact that the location of satellite footprints varies over time (i.e., the footprint of today's satellite overpass is not exactly the same as the footprint of the next overpass). Su et al. (2015) and Massari et al. (2017) did not have this issue as their products were all gridded.

**My reply:**

**Fine, but this means that the stations considered at different time steps will vary in your evaluation from time to time? Can you provide more details on this? If so this must be specified.**

Thanks for the comment. The stations used do differ depending on the time step and product, which we explicitly mention in the paper (Section 2.6): *"The final number of R, $R_{hi}$, and $R_{lo}$ values thus varied depending on the product."* This is, however, unavoidable, as the different products have different spatio-temporal coverages. Note that we also explicitly list the number of observations for each product in Figures 2 and 3. The satellite products without SWI filter, in particular, will use different sets of observations because of the different satellite overpass times. This is probably the case for every satellite soil moisture product evaluation to date and does not affect the robustness of the results.

4. Retaining only the *in situ* sensors with the best performance may paint an overly rosy picture of the products.

**My reply:**

**I do not fully agree with this. For the same temporal stability issue described above, the stations with the best performance are potentially the ones more representative of the spatial mean related to the domain of the satellite footprint. So it is not unfair to consider them but, to my opinion, it would be the best thing to do.**

We do not think it is always the case that the stations with the best performance are *"the ones more representative of the spatial mean related to the domain of the satellite footprint."* Regardless, it is not possible to implement this change; we have many different products with different footprint and grid-cells size, so which one should we choose? Should we choose a different measurement for each satellite product and time step and grid-cell? We believe this would make the evaluation extremely complicated, confusing, and irreproducible.

Vachaud, G., Passerat de Silans, A., Balabanis, P., & Vauclin, M. (1985). Temporal stability of spatially measured soil water probability density function. *Soil Science Society of America Journal*, *49*(4), 822-828.

**5) MAJOR**

**Old comment 3.**

**The overall methodology needs to be strongly improved and detailed as many aspects are not clear and/or not well discussed and justified:**

a. **The evaluation is carried by considering the temporal dynamic which is fine for the considerations done in the paper and from previous literature (see Koster et al. 2009), however, it is not clear how the evaluation at 3 hour resolution is done for satellite data with a revisit time larger than 1 day (e.g. SMAP, SMOS) and for model forced with rainfall with daily resolution. This must be clarified.**

*Reply by the authors:*

*We agree and have added the following text to explain this more clearly in the revised manuscript: "For the satellite products without SWI filter, we matched the instantaneous soil moisture retrievals with coincident 3-hourly in situ measurements to compute the R values."*

**My reply:**

**Thanks, this is clearer now for products where the Exponential filter was not applied. However, where the Exponential filter has been applied please refer to my comment 1 above.**

We are glad things are clearer now. Please see our earlier response regarding the exponential filter.

**6) MODERATE/MAJOR**

**Old Comment:**

**"T was set to 5 days for all products, as the performance did not change markedly using different values, as also reported in previous studies". The application of the exponential filter with a constant parameter T=5 days might be not appropriate for all the satellite products as the different products have a different vertical support. Since the calibration was carried out for the model why T was not calibrated also for the satellite products?**

*We strongly considered optimizing the time lag constant Tfor each product in the revised manuscript but in the end decided against this for two main reasons. First, we did not want to deviate too much from the original data because we want to make statements about the accuracy of the original data, not a post-processed product. Secondly, we did not want to give the satellite products an unfair advantage compared to the uncalibrated models, which would likely also benefit from the application of the SWI filter (though likely not as much).*

**My replies:**

*"First, we did not want to deviate too much from the original data because we want to make statements about the accuracy of the original data, not a post-processed product."*

**I think that downscaling satellite time series at 3-hourly resolution (from original revisit time of more than one day) by the application of the exponential filter (it does not matter whether T is 5, 7 or 3 days) already provides a strongly post-processed product.**

We appreciate the comment but respectfully disagree. Unlike the model products, the satellite products inevitably have gaps in their time series due to the limited number of satellite overpasses. In general, users of satellite soil moisture products will choose the observation nearest to the time of interest and thus implicitly apply a nearest neighbor filter. Since the exponential filter is very similar to the nearest neighbor filter, we do not agree that the products with exponential filter represent *"strongly post-processed products"*.

*"Secondly, we did not want to give the satellite products an unfair advantage compared to the uncalibrated models, which would likely also benefit from the application of the SWI filter (though likely not as much)."*

**R: Well, HBV is calibrated with 7 parameters on the 177 stations so I do not see limitations on doing the same calibration of the exponential filter with one single parameter (which, in its original formulation, is itself a conceptual approach to obtain root zone soil moisture). That is, the 177 calibration stations could be used to calibrate the parameter *T* which best fits observations in terms of correlation.**

Conceptual models like HBV do not have parameters with a clear physical meaning and therefore cannot be applied without some sort of calibration. This is not the case for the satellite products, which provide meaningful soil moisture values directly out of the box, without any optimization.

We could definitely have used an optimized *T* for each satellite product, but we decided to use a constant because, as stated in the manuscript, the results are fairly insensitive to the specific value of *T*, so an optimization would not have changed the results much (if at all).

The calibration of HBV was carried out because the model cannot be run without calibration, as it is a conceptual model with parameters that do not represent physical properties of the land surface. Note that we added the following regarding the generalization of the performance of the calibrated models to Section 3.9: *"The calibrated models (HBV and the Catchment model underlying SMAPL4) may, however, perform slightly worse in regions with climatic and physiographic conditions dissimilar to the in situ sensors used for calibration (but likely still better than the uncalibrated models)."*

**R:    remove the** *"but likely still better than the uncalibrated models"* **as it is not demonstrated or provide a reference to validate this statement.**

We consider it highly unlikely that the clear benefit of the calibration, demonstrated in the present study using completely independent *in situ* sensors, will disappear for *in situ* sensors in other regions. We are therefore not in favor of removing this accurate and informative statement.

**7) MODERATE/MAJOR**

**Old comment 6.**

**"The satellite products provided the least reliable soil moisture estimates and exhibited the largest regional performance differences on average, whereas the models with satellite data assimilation provided the most reliable soil moisture estimates and exhibited the smallest regional performance differences on average.". I think the authors should highlight again here that this result is expected given the high density gauge observations used in the study area. Highlighting this is very important as for instance ground validation conducted in data-rich areas does not adequately reflect the added values of satellite observations (Dong et al. 2019).**

Reply by the authors:

*Thanks for the comment. Even when excluding the three models with data assimilation using gauge-corrected precipitation forcings (GLEAM, SMAPL4, HBV-MSWEP+SMAPL3E), the remaining three models with data assimilation (ERA5, HBV-ERA5+SMAPL3E, and HBV-IMERG+SMAPL3E) still provide more reliable soil moisture estimates and smaller regional performance differences on average. This conclusion is thus not simply attributable to the inclusion of gauge observations in some of the precipitation forcings.*

**My Reply:**

**All ERA5 runs contain gauge precipitation in the US where most of the stations are located, so in practice, only HBV-IMERG+SMAPL3E (which is a calibrated product) has in theory no gauge information in it.**

Thank you for the comment. As explained earlier in this response, ERA5 cannot be considered a gauge-corrected product in the conventional sense.

**8) MODERATE**

**Old comment:**

**Line 24 pag. 12, "First, ESA-CCISWI incorporates ASCAT, which performed less well in the present evaluation, whereas". This cannot be a reason if the integration is "optimal" as the different parent products are weighed according to their relative performance. So the second one is more likely the reason. Please rephrase or justify with more solid arguments.**

*Reply of the authors:*

*The reviewer is right in theory; as discussed earlier in our response, given the difficulty of satisfying all triple collocation assumptions, our merging approach is unlikely to be fully "optimal," and we did not claim it was. For this reason, the inclusion of a product of lower quality results in a performance degradation. As mentioned before, we have added the following statement to the preceding paragraph to highlight this: "Triple collocation-based merging techniques rely on several assumptions (linearity, stationarity, error orthogonality, and zero cross-correlation; Gruber et al., 2016) which are generally difficult to fully satisfy in practice, affecting the optimality of the merging procedure."*

**My reply:**

**I think that ESA-CCI contains so many products and the merging procedure so complex that it is impossible to affirm that the guilty is one product rather than another one. ESA-CCI contains also SMOS which in Figure 2 is worse/equal to ASCAT but I do not feel to say the guilty is SMOS. Please revise this sentence or provide a more solid argument to state that.**

We agree and have revised the discussion of MeMo versus ESA-CCI as suggested by the reviewer. The discussion now reads as follows: *"We speculate that the better overall performance of MeMo compared to ESA-CCI$_{SWI}$ (Figs. 3a, 4, and 5) may be, at least partly, because ESA-CCI$_{SWI}$ incorporates ASCAT, which performed less well in the present evaluation, whereas MeMo incorporates SMAPL3E$_{SWI}$, which performed best among the single-sensor products (Figs. 3a and 4)."* We removed the second explanation provided in the original manuscript that ESA-CCI performed less because it uses only the soil moisture estimate from the 'best' sensor each day as this was incorrect. Note that the main developer of ESA-CCI is also co-author of the present study. Thanks for this comment.

**9) MODERATE**

**Old comment:**

**Line 3 pag. 13. "and satellite-based GPCP V1.3 Daily Analysis (Huffman et al., 2001)" How a daily rainfall can provide 3-hourly estimates?**

Answer by the authors:

Good question. This is explained in Section 2.1: *"Since the evaluation was performed at a 3-hourly resolution, we downscaled the two products with a daily temporal resolution (VIC-PGF and GLEAM) to a 3-hourly resolution using nearest neighbor resampling."* We realize that this is not ideal, but there was no other solution.

**My reply:**

**Can you clarify it better? Do you downscale GPCP daily to 3 hourly data? So the daily value is divided by 8 to have consistent daily accumulations?**

Apologies, we misread the question. Daily GPCP precipitation was disaggregated to 3-hourly using reanalysis data. The disaggregated precipitation data was subsequently used to force the Noah land surface model among others. For more information, see Rui et al. (2020).

In case the nearest neighbour resampling procedure used to downscale the daily VIC-PGF and GLEAM data to 3-hourly in our study was not clear, we added the following text: *"(resulting in replication of the daily value for all 3-hourly periods on each day)"*.

**10) MODERATE**

**I think it is important to provide some plots of the time series for instance for one/two locations (to put at least in the supplementary information) to better visualize the impact of the downscaling procedure and the visual comparison between the products.**

Thank you for the suggestion. We have added the following figure (Fig. 1 of the revised manuscript) to illustrate the exponential SWI filter:

[revised manuscript text omitted]